# Adaptive tuning of mutation rates allows fast response to lethal stress in *Escherichia coli*

Toon Swings[1], Bram Van den Bergh[1], Sander Wuyts[1], Eline Oeyen[1], Karin Voordeckers[1,2], Kevin J Verstrepen[1,2], Maarten Fauvart[1,3], Natalie Verstraeten[1], Jan Michiels[1]*

[1]Centre of Microbial and Plant Genetics, KU Leuven - University of Leuven, Leuven, Belgium; [2]VIB Laboratory for Genetics and Genomics, Vlaams Instituut voor Biotechnologie, Leuven, Belgium; [3]Smart Systems and Emerging Technologies Unit, Imec (Interuniversity Micro-Electronics Centre), Leuven, Belgium

**Abstract** While specific mutations allow organisms to adapt to stressful environments, most changes in an organism's DNA negatively impact fitness. The mutation rate is therefore strictly regulated and often considered a slowly-evolving parameter. In contrast, we demonstrate an unexpected flexibility in cellular mutation rates as a response to changes in selective pressure. We show that hypermutation independently evolves when different *Escherichia coli* cultures adapt to high ethanol stress. Furthermore, hypermutator states are transitory and repeatedly alternate with decreases in mutation rate. Specifically, population mutation rates rise when cells experience higher stress and decline again once cells are adapted. Interestingly, we identified cellular mortality as the major force driving the quick evolution of mutation rates. Together, these findings show how organisms balance robustness and evolvability and help explain the prevalence of hypermutation in various settings, ranging from emergence of antibiotic resistance in microbes to cancer relapses upon chemotherapy.

*For correspondence: jan. michiels@kuleuven.be

Competing interests: The authors declare that no competing interests exist.

## Introduction

Theory predicts that the optimal mutation rate depends on several different factors, including genome size and effective population size. For example, unicellular organisms, such as viruses and bacteria, exhibit per-base mutation rates that are inversely correlated with their population and genome sizes, whereas multicellular organisms with much larger genomes and smaller effective population sizes may have higher per-base mutation rates (*Lynch, 2010*; *Lynch et al., 2016*). While mutations are necessary to adapt to new and stressful environments (*Barrick and Lenski, 2013*; *Wiser et al., 2013*), random changes in an organism's DNA are rarely beneficial, but more often neutral or slightly deleterious (*Elena and Lenski, 2003a*; *Perfeito et al., 2007*; *Eyre-Walker and Keightley, 2007*). Consequently, mutation rates are strictly balanced by the trade-off between the need for mutations to adapt and the concomitant increase in genetic load (*Sniegowski and Raynes, 2013*; *Lynch, 2011*; *Wielgoss et al., 2013*; *Desai and Fisher, 2011*). This trade-off between adaptability and adaptedness is believed to be responsible for the low genomic mutation rates usually observed in organisms, while a further decrease in mutation rate is restricted by the energy needed to increase and maintain high replication fidelity (*de Visser, 2002*; *Sung et al., 2016*; *Sniegowski et al., 2000*; *Ram and Hadany, 2014*). Furthermore, the power of random drift will limit selection on even lower mutation rates when additional increases in replication fidelity are

**eLife digest** A cell's DNA can acquire errors over the course of its lifetime. These errors, known as mutations, are often harmful and can cripple the cell. However, some mutations are needed to enable a cell or organism to adapt to changes in its environment. Since there is a trade-off between acquiring beneficial mutations versus harmful ones, cells carefully balance how often they acquire new mutations.

Cells have several mechanisms that limit the number of mutations by correcting errors in DNA. Occasionally these repair mechanisms may fail so that a small number of cells in a population accumulate mutations more quickly than other cells. This process is known as "hypermutation" and it enables some cells to rapidly adapt to changing conditions in order to avoid the entire population from becoming extinct.

So far, studies on hypermutation have largely been carried out under conditions that are mildly stressful to the cells, which only cause low frequency of hypermutation. However, little is known about the role of this process in cells under near-lethal levels of stress, for example, when drugs target bacteria or cancer cells in the human body.

Swings *et al.* studied hypermutation in populations of a bacterium called *Escherichia coli* exposed to levels of alcohol that cause the bacteria to experience extreme stress. The experiments show that hypermutation occurs rapidly in these conditions and is essential for bacteria to adapt to the level of alcohol and avoid extinction. Populations of bacteria in which hypermutation did not occur were unable to develop tolerance to the alcohol and perished. Further experiments show that an individual population of bacteria can alter the rate of mutation (that is, how often new mutations arise) several times as a result of changing stress levels.

The findings of Swings *et al.* suggest that hypermutation can rapidly arise in populations of cells that are experiencing extreme stress. Therefore, this process may pose a serious risk in the development of drug resistant bacteria and cancer cells. In the future, developing new drugs that target hypermutation may help to fight bacterial infections and cancer.

insufficiently advantageous. Due to this drift-barrier, mutation rates are believed to evolve to an equilibrium where the strength of selection matches the power of drift (*Lynch et al., 2016*).

An organism's cellular mutation rate is generally considered to be near-constant (*Lynch, 2010*; *Drake, 1991*), yet the optimal mutation rate has been reported to depend on the environment (*Elena and de Visser, 2003b*; *Rando and Verstrepen, 2007*). Wild-type bacteria grown under optimal conditions typically have low mutation rates in the order of $10^{-3}$ mutations per genome per generation (*Lee et al., 2012*; *Drake et al., 1998*). Under these conditions, hypermutators with weakly (10-fold) or strongly (100–10.000-fold) increased mutation rates occur only sporadically (*de Visser, 2002*; *Marinus, 2010*). Despite low frequencies of hypermutators in laboratory populations (*Boe et al., 2000*), a much higher prevalence is observed in natural bacterial populations (*Gross and Siegel, 1981*; *Hall and Henderson-Begg, 2006*), such as clinical isolates of pathogenic *E. coli* (*Matic et al., 1997*; *Denamur et al., 2002*; *LeClerc et al., 1996*), *Pseudomonas aeruginosa* (*Marvig et al., 2013*, *2015*; *Oliver, 2015*; *Ferroni et al., 2009*), *Salmonella* (*LeClerc et al., 1996*), *Staphylococcus aureus* (*Iguchi et al., 2016*) among others (*Negri et al., 2002*; *Gould et al., 2007*; *Rajanna et al., 2013*) and in nearly all *A. baumannii* strains adapting to severe tigecycline stress (*Hammerstrom et al., 2015*). In addition, high frequency of hypermutation is also documented in eukaryotic pathogens including the malaria-causing parasite *Plasmodium falciparum* (*Lee and Fidock, 2016*; *Gupta et al., 2016*) and the fungal pathogen *Candida glabrata* (*Healey et al., 2016*). Moreover, hypermutation plays an important role in cancer development and proliferation, as it helps to overcome different barriers to tumor progression (*Wang et al., 2016*; *Bielas et al., 2006*; *Roberts and Gordenin, 2014*). These observations suggest the natural occurrence of situations in which higher mutation rates confer a selectable advantage. This is especially obvious in harsh environments, where near-lethal stress requires swift adaptation of at least some individuals to avoid complete extinction of the population (*Bell and Gonzalez, 2011*). Adaptation sufficiently rapid to save a population from extinction is called evolutionary rescue. This phenomenon is widely studied

in the light of climate change and adaptation of declining populations to new environments (*Lindsey et al., 2013*). It occurs when a population under stress lacks sufficient phenotypic plasticity and can only avoid extinction through genetic change (*Gonzalez et al., 2013*). Evolutionary rescue depends on different factors such as the population size, genome size, mutation rate, degree of environmental change and history of the stress (*Gonzalez et al., 2013*; *Gonzalez and Bell, 2013*). By increasing the supply of mutations, hypermutation might also be crucial to enable evolutionary rescue for populations under near-lethal stress.

Despite the high prevalence of hypermutation in clinical settings, current knowledge is lacking on the long-term fate of mutators and their specific role in survival under near-lethal stress conditions. Previous studies exploring the costs and benefits of mutators mostly focused on mild stresses. Both experimental evidence and theory show that mutators can readily increase in frequency in a population through second-order selection. In this case, a mutator hitchhikes along with a sporadically occurring, beneficial mutation that thrives under natural selection (*Gentile et al., 2011*; *Woods et al., 2011*; *Giraud et al., 2001*; *Shaver et al., 2002*; *Mao et al., 1997*; *Sniegowski et al., 1997*). This process relies on different elements, such as initial mutator frequency (*Tenaillon et al., 1999*), the relative timing of the emergence of one or multiple beneficial mutations (*Tanaka et al., 2003*), the degree of environmental change or selection strength (*Mao et al., 1997*; *Pal et al., 2007*), the mutational spectrum (*Couce et al., 2013*) and the strength of the specific mutator (*Loh et al., 2010*). Although hypermutation can readily spread in a population by means of hitchhiking when adaptation is required, long-term evolution experiments also show selection against hypermutation (*Lynch, 2011*; *Tröbner and Piechocki, 1984*). These results demonstrate that the actual mutation rate of a population is prone to change by evolution. However, our current knowledge on the long-term dynamics of hypermutation and the mechanisms underlying changes in mutation rate remains fragmentary. Specifically, conditions under which the spread of mutators is inhibited or the increased mutation rate is reversed, remain largely unexplored (*Raynes and Sniegowski, 2014*).

The aim of the current study was to better understand the dynamics of hypermutation under near-lethal, complex stress. Therefore, we used *E. coli* exposed to high ethanol stress as a model system (*Goodarzi et al., 2010*; *Nicolaou et al., 2012*). Here, multiple mutations epistatically interact and diverse evolutionary trajectories can lead to adaptation to high ethanol concentrations (*Voordeckers et al., 2015*). In our study, we found an unexpected flexibility in cellular mutation rates as a response to changes in selective pressure. First, we used a defined collection of mutators with distinct mutation rates to identify a range of optimal mutation rates to enable rapid growth under high ethanol stress. Next, experimental evolution revealed an essential role for hypermutation for de novo adaptation to high ethanol stress. While hypermutation quickly and recurrently arose concurrent with increases in ethanol concentrations, mutation rates rapidly declined again once cells were adapted to the stress. Interestingly, we identified cellular mortality as the major force that drives fast evolution of mutation rates. In summary, our results shed new light on the dynamics of mutation rate evolution and help explain why maintaining high mutation rates is limited in time.

## Results

### Hypermutation enables rapid growth under high ethanol stress

Little is known on the role of hypermutators under complex, near-lethal stress conditions. In these conditions, growth rates are low and the probability to accumulate an adaptive mutation is strongly limited. We postulate that mutator mutants yield variable benefits under these conditions, depending on their mutation rates. To verify this hypothesis, a collection of *E. coli* mutants displaying a range of mutation rates (*Figure 1—figure supplement 1*) was grown in 5% EtOH. At this concentration, ethanol almost completely inhibits growth and drastically reduces the carrying capacity of a wild-type culture, indicating extreme stress (*Figure 1—figure supplement 2*).

Growth rate and lag time reflect the fitness of a strain in a specific environment (*Stepanyan et al., 2015*; *Hammerschmidt et al., 2014*). These growth parameters are contingent upon the initial population size. On the one hand, the effect of a rare beneficial mutation on growth rate and lag time is mitigated by a large initial population size (*Mao et al., 1997*). On the other hand, the effect of a beneficial mutation on the growth dynamics is amplified by a small initial population size, as this limits the generation of beneficial mutants. Therefore, we tested growth of wild-

type and mutator strains in the presence of 5% EtOH both for a small ($10^4$ CFU/ml) and a large ($10^7$ CFU/ml) population size. In the latter condition, we observed strongly overlapping growth curves. Small initial population sizes, however, led to highly dispersed growth curves, pointing to an important contribution of mutation rates to the adaptive capacity under ethanol stress (**Figure 1a**). Surprisingly, large initial populations lead to a lower yield compared to small initial population sizes. The growth from the small inoculum is likely driven by adaptive mutations, while the effect of a beneficial mutation might be mitigated when starting with a large inoculum. Moreover, we expect that a mutant occurring in case of a small initial inoculum size will have more time to manifest ($log_2(dilution\ factor : 100\ 000) = \pm16.61$ generations), compared to the mutant occurring in case of a large initial population size ($log_2(100) = \pm6.67$ generations), possibly leading to the observed higher yield.

The lag times calculated from the growth curves reveal a window of beneficial mutation rates for growth under 5% EtOH (**Figure 1—figure supplement 3**). Strikingly, 2- to 70-fold increased mutation rates (*e.g.* in a Δ*mutS* mutant) are more advantageous under these conditions than mutants with lower or higher mutation rates (*e.g.* in Δ*xthA* or Δ*dnaQ* mutants, respectively). In the absence of ethanol, we did not observe differences in lag time among the wild type and mutator mutants. This suggests a crucial role for hypermutation for rapid growth under near-lethal stress by supplying the population more rapidly with (a combination of) beneficial mutations (**Figure 1—figure supplement 3**).

Growth rates, in turn, when calculated from a large initial density of $10^7$ cells per ml, did not differ significantly in the presence of 5% EtOH, suggesting no direct fitness effect for hypermutation in high ethanol conditions (**Figure 1b**). However, when growth rates were determined for each mutant starting from small initial population sizes, we observed higher growth rates for all mutator mutants relative to the wild type, demonstrating the emergence of adaptive mutations (**Figure 1b**). These data indicate that the advantage of hypermutation under ethanol stress can be attributed mainly to second-order selection, following the beneficial effects of novel mutations relative to possible direct effects of the mutator mutation itself. To corroborate these results, we determined relative fitness (**Van den Bergh et al., 2016**) from direct competition experiments between the wild type and mutants with contrasting mutation rates (Δ*mutM*, Δ*mutS* and Δ*dnaQ*). These tests demonstrate the fitness advantage of hypermutation under 5% EtOH. In accordance with the data of the lag time, the relative fitness compared to the wild type is high for the Δ*mutS* strain while it does not differ from one for the Δ*mutM* and Δ*dnaQ* mutants (**Figure 1—figure supplement 4**). These results confirm that the advantage of hypermutation under near-lethal stress can be attributed to the rapid emergence of beneficial mutations, enabling fast adaptation to avoid extinction.

## Long-term adaptation to high ethanol stress in *E. coli* is contingent upon hypermutation

Our results suggest an essential role for hypermutation in evolution under near-lethal stress. To further extend these observations to a wild-type population, we set up a long-term evolution experiment aimed at adapting *E. coli* to high percentages of ethanol. We serially transferred 20 parallel *E. coli* lines founded by a non-mutator ancestor for approximately two years (more than 500 generations). To maintain near-lethal ethanol concentrations throughout the adaption process, populations were incubated in gradually increasing ethanol concentrations (**Figure 2a**; **Figure 2—figure supplement 1**). Although ethanol tolerance increased in all populations, only eight out of 20 lines developed tolerance to very high (7% or more) ethanol concentrations (**Figure 2b**), while the other 12 lines recurrently died out and only developed tolerance to relatively low ethanol concentrations (6% or lower). These results suggest the presence of a critical factor inherent to those eight lines that underlies their increased ethanol tolerance.

To explore the fundamental difference between high ethanol tolerant lines and low ethanol tolerant lines, we used fluctuation assays to determine the population mutation rate. The results clearly show that lines can be divided in two groups with mutation rates either higher or lower than the wild-type mutation rate. This subdivision perfectly corresponds to the difference in ethanol-tolerance levels (**Figure 3a**). In conclusion, even though mutator mutants occur spontaneously in the population, these data suggest that hypermutation underlies adaptation to high ethanol levels in such a way that only lines with a higher mutation rate than the wild-type mutation rate are able to evolve high ethanol tolerance (**Figure 3b**).

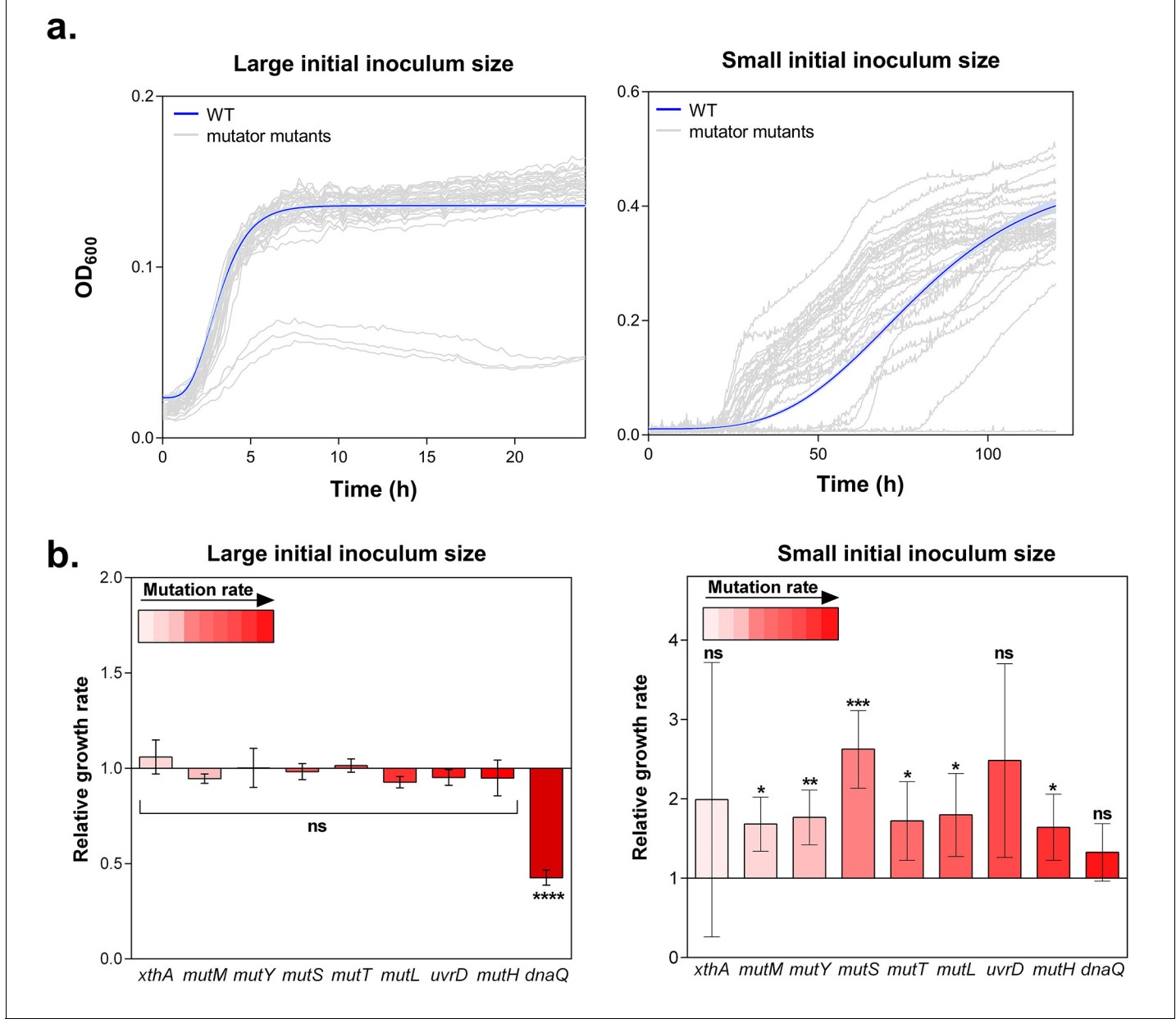

**Figure 1.** Hypermutation favors growth in high EtOH stress through generation of beneficial mutations. (a) In the left panel, the large initial population size ($10^7$ CFU/ml) mitigates the effect on growth of emerging beneficial mutations. Growth curves of both wild type and mutators are overlapping except for all replicates of the Δ*dnaQ* mutant. In the right panel, we observed highly dispersed growth curves. The effect of a beneficial mutation manifests itself due to the small initial population size ($10^4$ CFU/ml). The blue line and shading represents the sigmoidal fit of the wild-type growth curves (n = 3, fit using Gompertz equation with 95% c.i. (shading), see *Equation 1* in Materials and methods section), while the grey lines represent growth curve of separate replicates for each mutator mutant (b) Growth rates of all strains in the presence of 5% EtOH were measured both starting from a large initial population size of $10^7$ cells per ml (left) and a small initial population size of $10^4$ cells per ml (right). No significant difference was observed between the growth rates of the wild type and mutants in the case of a large starting population, indicating no direct fitness effect caused by the deletion of mutator genes (except for the Δ*dnaQ* mutant) (mean ± s.d., n = 3, repeated measures ANOVA with post hoc Dunnett correction, ****p<0.001). When starting from a small initial population, growth rates of all mutator mutants increased compared to the wild type (mean ± s.d., n = 3, two-sided Student's t-test, *p<0.1; **p<0.05; ***p<0.01; ns: not significant), indicative of the occurrence of adaptive mutations as an indirect benefit for hypermutation under complex, near-lethal stress.

The following figure supplements are available for figure 1:

**Figure supplement 1.** Deletion of selected mutator genes causes increased mutation rates under normal growth conditions.

*Figure 1 continued on next page*

*Figure 1 continued*

**Figure supplement 2.** 5% EtOH mimics near-lethal stress and leads to a severe decrease in growth rate and a decrease in carrying capacity.

**Figure supplement 3.** Lag times reveal a window of optimal mutation rates for growth in the presence of 5% EtOH.

**Figure supplement 4.** Relative fitness W associated with different mutation rate variants reveals an advantage for the Δ*mutS* mutator under EtOH stress.

Mixed pools and one characterized clone of the endpoints of all high tolerant and two low tolerant lines were subjected to whole genome resequencing. In all lines that developed high ethanol tolerance, a prominently high number of mutations was present, compared to the number of mutations in the low tolerant lines (*Figure 2—figure supplement 2*). This confirms the existence of a hypermutation phenotype in the highly tolerant lines. Furthermore, the observed mutational spectrum reveals the typical pattern expected for methyl-directed mismatch repair (MMR) mutators, in which case transitions are strongly favored over transversions (*Schaaper and Dunn, 1987*) (*Figure 2—figure*

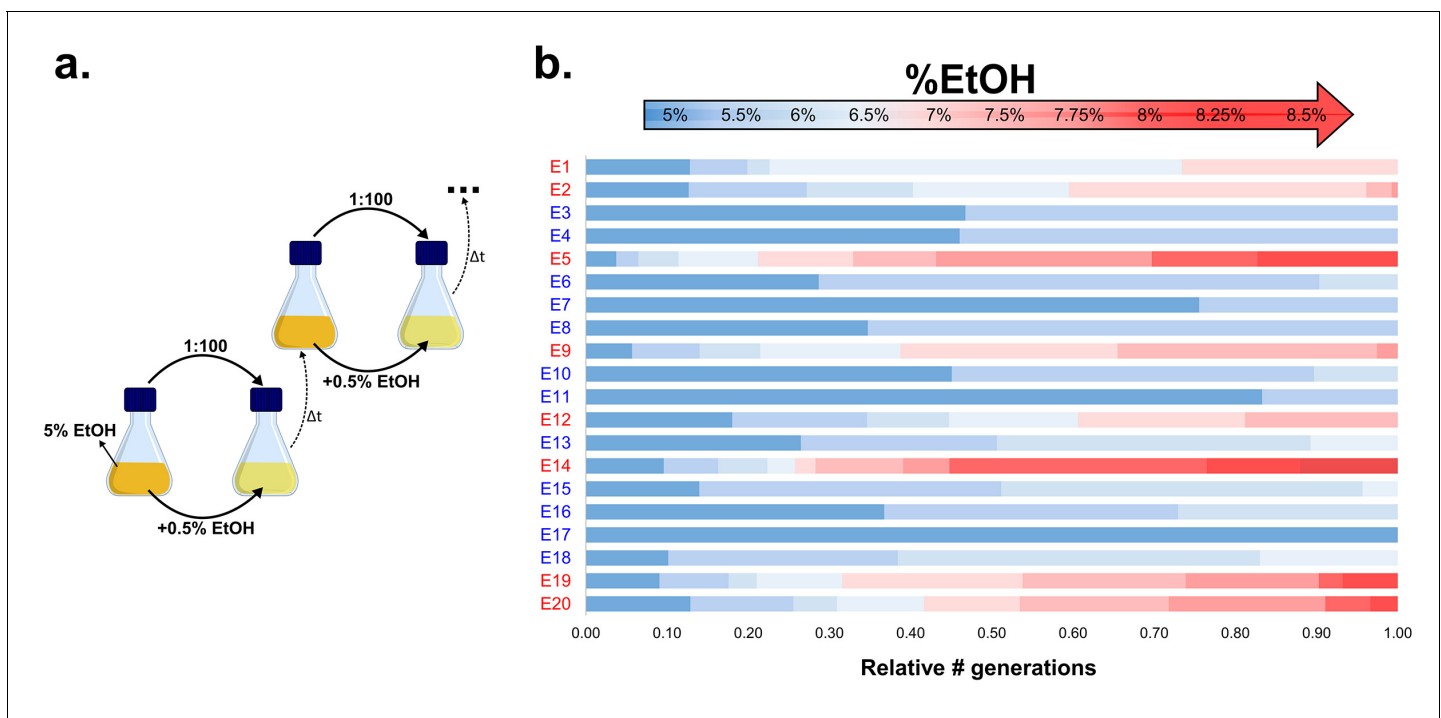

**Figure 2.** Experimental evolution of *E. coli* to increasing EtOH concentrations. (a) Setup of the evolution experiment with increasing percentage of EtOH. Initially, ancestral cells were grown in the presence of 5% EtOH, the condition that mimics near-lethal stress (*Figure 1—figure supplement 2*). Populations that grew until exponential phase were transferred to fresh medium while simultaneously increasing EtOH concentrations with 0.5% (for full details, see Materials and methods). (b) Evolutionary outcome of 20 independent parallel lines. Eight parallel lines evolved to high EtOH tolerance (shown in red). The other 12 lines were only able to acquire low EtOH-tolerance levels (shown in blue). For each line, the relative time (in generations) it spent growing on a certain percentage of EtOH is shown.

The following figure supplements are available for figure 2:

**Figure supplement 1.** Flowchart of the experimental evolution experiment to high EtOH tolerance in *E. coli*.

**Figure supplement 2.** The total number of mutations exceeds the number of fixed mutations in the population.

**Figure supplement 3.** The mutational spectrum of evolved EtOH-tolerant lines corresponds to the mutational spectrum of MMR mutators.

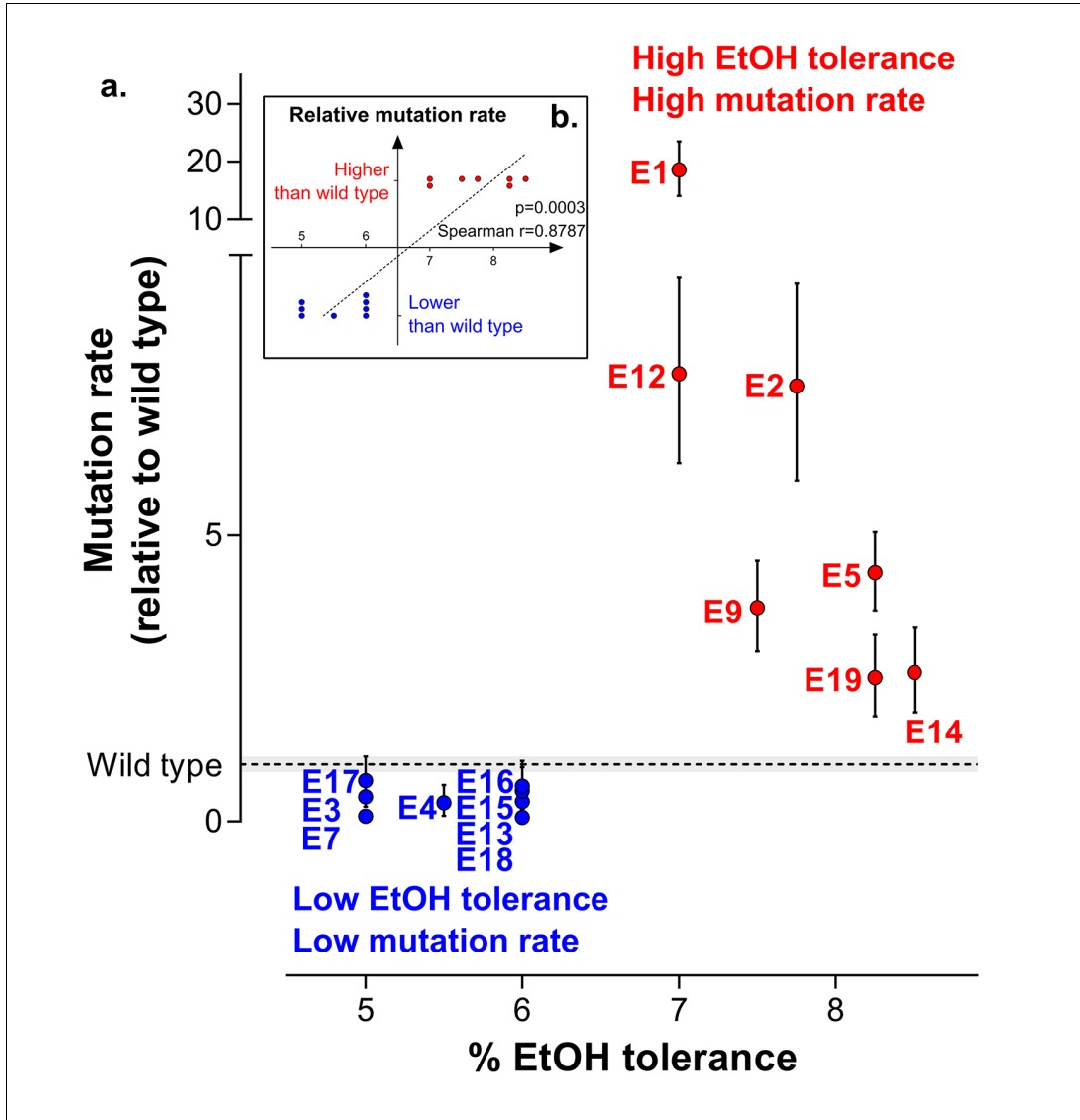

**Figure 3.** Increased mutation rate underlies evolution of high EtOH tolerance. (a) The population mutation rate of parallel evolved lines relative to the wild type mutation rate is shown (mean ± 95% c.i., see Materials and methods). Two different groups can clearly be distinguished according to the higher than wild-type ( 🔴 ) or lower than wild-type ( 🔵 ) mutation rate. This subdivision is in accordance with the difference in endpoint EtOH tolerance levels (*Figure 2b*). All mutation rates were significantly different from the wild type (p<0.001; two-sided Student's t-test on the absolute number of mutational events as calculated by FALCOR, assuming equal cell densities [see Materials and methods]) (b) For correlation analysis, all parallel lines were subdivided in two groups according to their higher or lower than wild type mutation rate. Spearman correlation analysis resulted in a highly significant positive correlation (p<0.001). Lines with a mutation rate lower or equal than the wild-type mutation rate are therefore correlated with lower ethanol tolerance, whereas lines with a higher mutation rate than the wild-type mutation rate are correlated with high ethanol tolerance. In conclusion, these data suggest that hypermutation is necessary for adaptation to high EtOH stress.

supplement 3a). We therefore scanned the population sequence data for mutations in genes involved in DNA replication and repair. We found different *mutS* mutations in six out of eight highly ethanol-tolerant lines (lines E1, 3, 9, 12, 19 and 20). In the remaining highly tolerant lines E5 and E14, as well as in E9, we found fixed mutations in *mutL*, while line E9 additionally acquired a mutation in *mutH*, strongly suggesting a deficient MMR pathway (*mutS,L,H*) as the main cause of increased mutation rates under near-lethal ethanol stress (*Figure 2—figure supplement 3b*). In

addition, mutations in other possible mutator genes (*xthA*, *mutY* and *uvrD*) appeared later on in evolution (i.e. after the occurrence of the MMR mutations).

To confirm the role of MMR, we evaluated the selective advantage of a specific *mutS* point mutation (G100A) originating from one of the highly-tolerant lines (E1), in the presence of 5% EtOH. This mutation is located near the mismatch recognition site of MutS and causes an approximately 10-fold increase in mutation rate (*Figure 4a*). The *mutS*$_{G100A}$ and Δ*mutS* strains were competed directly against the wild-type strain under 5% EtOH (*Figure 4—figure supplement 1a*). The frequency of both mutator strains increased for all initial cell ratios, resulting in rapid fixation in the population after one growth cycle (*Figure 4b*). Additionally, we observed a distinct pattern for both mutation rate variants. Calculation of the relative fitness (*Van den Bergh et al., 2016*) (*Figure 4—figure supplement 1b*) confirms our previous results (*Figure 1*, *Figure 1—figure supplement 3*) showing that a Δ*mutS* mutant is more fit under 5% EtOH stress as compared to variants with a lower mutation rate, such as the *mutS*$_{G100A}$ mutant.

Even though all mutator mutations were 100% fixed in the population, we were able to detect a vast amount of low frequency mutations present in the population (*Figure 2—figure supplement 2*). The graph reveals the difference between total amount of mutations (>10% frequency) and number of 'fixed' mutations (>75% frequency). Only a fraction of the variants are fixed in the hypermutating lines, suggesting a complex population structure with different subpopulations (*Pulido-Tamayo et al., 2015*). Consequently, the population mutation rate will reflect the average genomic mutation rate of the entire population, containing different subpopulation that possibly display

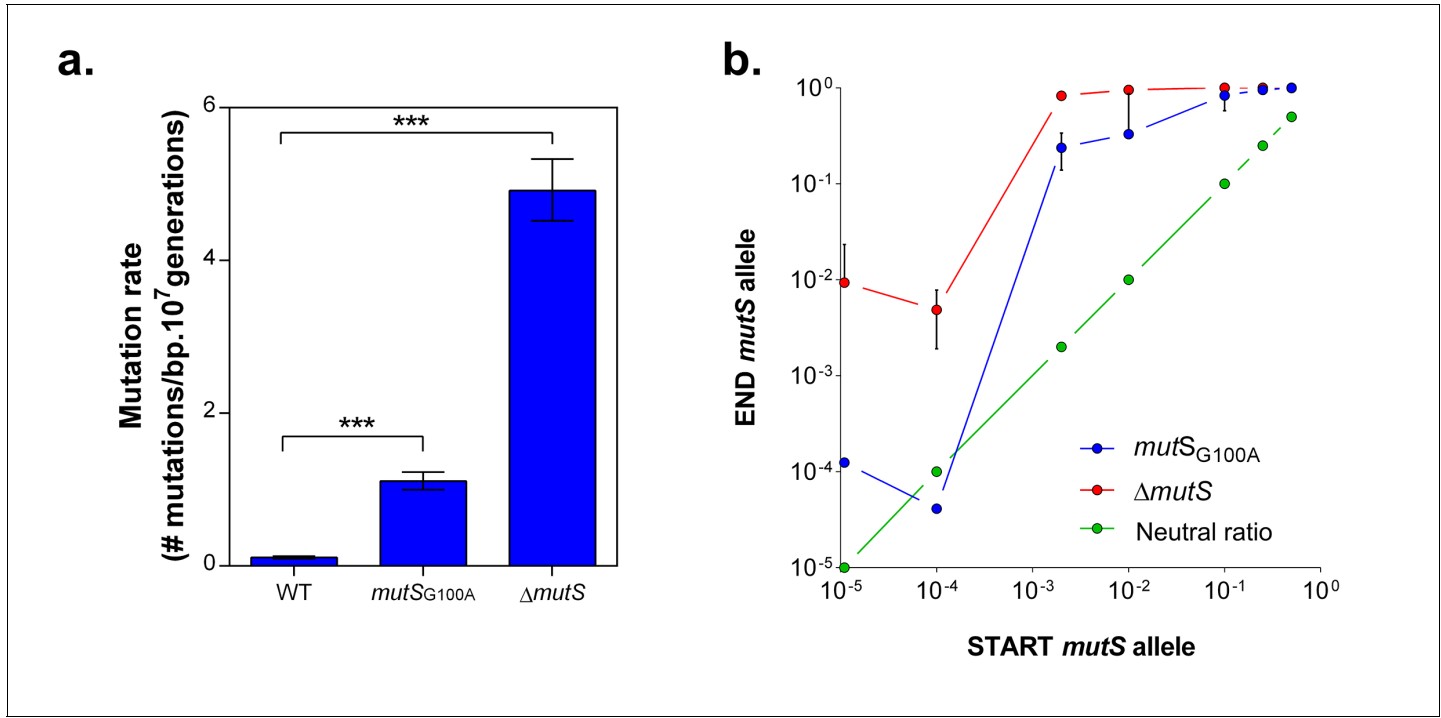

**Figure 4.** *mutS* mutators are able to outcompete wild-type cells in direct competition under 5% EtOH, irrespective of the mutator population size. (a) The specific G100A point mutation in *mutS* was introduced in a wild-type background. This point mutation confers a significantly increased mutation rate compared to the wild type strain (mean ± 95% c.i., \*\*\*p<0.001, see Materials and methods). The mutation rate of the *mutS*$_{G100A}$ strain is lower than the mutation rate of the clean Δ*mutS* knockout mutant (1.1127 vs 4.1971 mutations per bp per $10^7$ generations). (b) The green line (●) represents the expected ratio if there is no fitness effect. The red line (●) gives the results for the Δ*mutS* mutant and the blue line (●) represents the results for the *mutS*$_{G100A}$ mutant (mean ± s.d., n = 3). For both mutants, an increase in fraction of mutators in the population was seen, showing the advantage of hypermutation under high EtOH stress.

The following figure supplement is available for figure 4:

**Figure supplement 1.** Setup *mutS* mutator competition experiment under 5% EtOH.

above- or below-average mutation rates. This may already explain the discrepancy between the 20-fold increased endpoint mutation rate of line E1 (*Figure 3a*) and the 10-fold increased clonal mutation rate caused by the $mutS_{G100A}$ mutation identified in that same line (*Figure 4a*). Furthermore, these data suggest that mutation rate can vary along with population structure throughout the evolution rather than being a fixed rate after the occurrence and spread of one mutator mutation.

## Dynamics in mutation rate underlie evolution to high ethanol tolerance

To understand dynamics of mutation rates, we analyzed the occurrence of variations in mutation rate during evolution towards high ethanol tolerance by measuring the genomic mutation rate of populations sampled at different time points during the evolution experiment. The results reveal a dynamic pattern of rapidly altering mutation rates (*Figure 5a*). Furthermore, there is a significant, positive correlation (p<0.05) between changes in ethanol tolerance and differences in mutation rate between two consecutive time points (*Figure 5b*). Next, we measured the mutation rate of two selected intermediate point (IM2 and IM3, *Figure 6a*) in the presence of 7% ethanol. The relative fold-change decrease in mutation rate between these two points was unaffected by the presence of ethanol compared to the ±7-fold change in the absence of ethanol, strongly suggesting that ethanol itself does not change the dynamics caused by differences in the genomic mutation rate (*Figure 5—figure supplement 1*). In conclusion, increased mutation rate co-occurs with increased ethanol tolerance, likely because mutator mutations hitchhike along with a beneficial mutation. This is in line with results from our previous competition experiments (*Figures 1*, *4*), showing that direct selection on adaptive mutations increases ethanol tolerance while the hitchhiking of mutator mutations increases the mutation rate in the population accordingly.

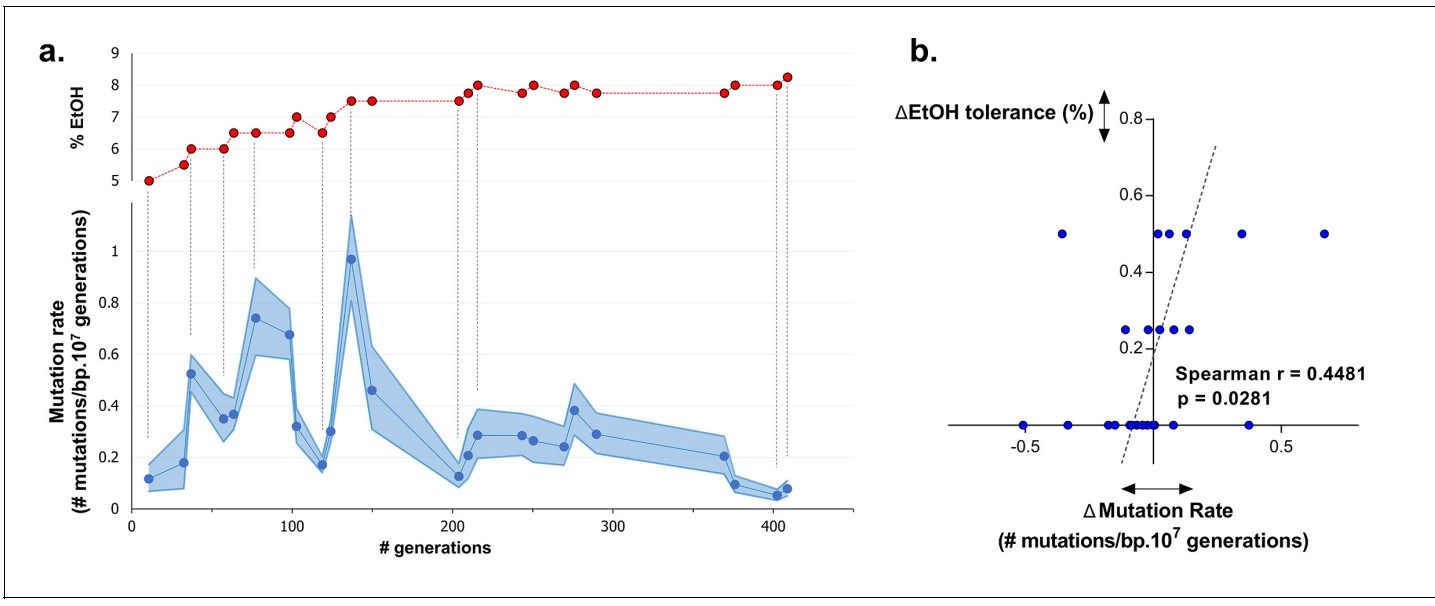

**Figure 5.** Dynamics in population mutation rate underlie evolution to high EtOH tolerance. (a) The genomic mutation rate in the absence of EtOH is shown for selected intermediate time points of line E5 ( ⬤ ) (mean ± 95% c.i. (blue shading), see Materials and methods). In the top graph, the EtOH tolerance associated with each time point is shown ( 🔴 ) and corresponding points in both graphs are connected by dashed lines. Increases in EtOH tolerance co-occur with increases in mutation rate, suggesting the hitchhiking of a mutator mutation with adaptive mutations conferring higher EtOH tolerance. During periods of constant EtOH exposure, mutation rates decline, suggesting that once a strain is adapted to a certain percentage of EtOH, high mutation rates become deleterious and selection acts to decrease the mutation rate. (b) The difference in mutation rate at consecutive time points and the difference in EtOH tolerance correlate positively (Spearman rank coefficient = 0.4481, p<0.05). The dashed line represents the linear regression through the data points.

The following figure supplement is available for figure 5:

**Figure supplement 1.** Dynamics in mutation rate during evolution are not affected by ethanol itself.

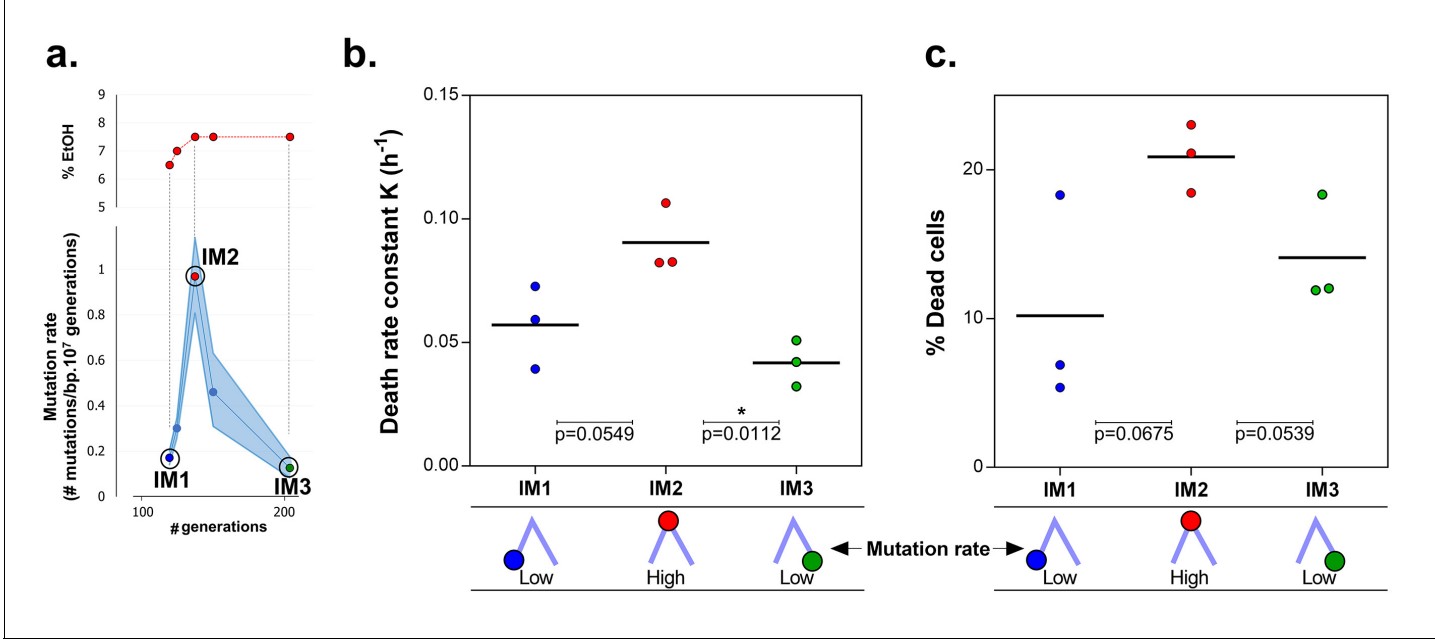

**Figure 6.** Mortality is the cost of hypermutation in evolved strains when high EtOH tolerance is reached. (a) The specified intermediate time points from high EtOH-tolerant line E5 were selected based on their different mutation rate (*Figure 5*). (b) The death rate constant for each intermediate point is shown (mean, n = 3, one-phase exponential decay fitting on the decrease in viable cell count, see Materials and methods, *Figure 6—figure supplement 1*). The death rate constants were statistically compared using a one-way ANOVA with post hoc Tukey correction (*p<0.05). (c) The percentage dead cells as determined by live-dead staining (see Materials and methods) is shown for each intermediate time point. The fractions of dead cells were statistically compared using a two-sided Student's t-test (mean, n = 3, *p<0.05). An increase in mutation rate coincides with both an increase in death rate constant and an increase in fraction of dead cells in the population. A subsequent decrease in mutation rate coincides with a decrease in death rate constant and fraction of dead cells. These data suggest that higher mortality is the cost of hypermutation when adaptation to a certain level of EtOH stress is achieved.

The following figure supplements are available for figure 6:

**Figure supplement 1.** Number of viable cells decreases more rapidly in highly EtOH-tolerant strains than in the wild type.

**Figure supplement 2.** The death rate is constant during exponential-phase growth of IM1 (*Figure 6*).

**Figure supplement 3.** Direct effects of acquired mutations possibly cause the discrepancy between calculated and observed number of selection rounds necessary for fixation of IM3.

## Cellular mortality is the underlying force driving evolution of mutation rates

Remarkably, the mutation rate decreases quickly in the long-term evolution experiment during periods when the concentration of ethanol is kept constant (*Figure 5a*). This fast decrease can either be explained by reversion of mutator mutations or by the accumulation of compensatory suppressor mutations (*Wielgoss et al., 2013*). We tested the former by targeted sequencing of the mutations that were acquired in the MMR genes in intermediate points before and after the decrease in mutation rate. No such reversions of mutator mutations were found. These results therefore suggest that suppressor mutations have accumulated in the ethanol-tolerant mutator lines. To unravel the benefit of a lower mutation rate and the cost of hypermutation when high ethanol tolerance is reached and thus selective pressure for ethanol tolerance is not further increased, we selected intermediate time points (IM1, IM2 and IM3) with contrasting mutation rates (*Figure 6a*). These intermediate points were used to determine cell viability under high ethanol stress and to extract relevant growth parameters by fitting a bacterial growth equation to the growth dynamics (See Materials and methods). Cell death was determined both by quantifying viable cells and by live-dead staining. Surprisingly,

all tested intermediate points showed a very fast decrease in viable cell count when entering the stationary phase (*Figure 6—figure supplement 1*). This decrease in viable cells is explained by a genuine increased death rate in the population since cells are in stationary phase. We fitted this decrease in viable cells with an exponential decay function and extracted the death rate constant (See Methods). The fitting data demonstrate an association between an increase in mutation rate and an increase in death rate constant (*Figure 6b*). In addition, death rates significantly decline as mutation rates decrease. These results were confirmed by live-dead staining and subsequent flow cytometry analysis (*Figure 6c*). Moreover, a constant death rate was measured throughout the growth cycle of the strain (*Figure 6—figure supplement 2*). Intermediate evolved strains with increased ethanol tolerance are thus characterized by high death rates which are dependent on the mutation rates. Likely, these strains have accumulated a high genetic load throughout the evolution experiment. Our data now suggest that a further buildup of genetic load and a higher chance to acquire a lethal mutation cause increased mortality, which results in a selective pressure per se. Strains with lower mutation rates resulting from compensatory mutations can increase their fitness due to decreased death rates. To corroborate these data, we used time to grow and optical density data from the initial evolution experiment to calculate the relative fitness of IM3 (low mutation rate) compared to IM2 (high mutation rate) in the presence of 7.5% ethanol. We used the relative fitness to calculate the theoretical number of selection rounds necessary for IM3 to fix in a population of IM2 and compared it to the actual number of selection rounds (*Figure 6—figure supplement 3*). The discrepancy between those two values suggests that fixation happened faster than theoretically possible given the calculated relative fitness. Additional mutations that occurred between IM2 and IM3 might affect the speed of selection. Further buildup of genetic load and a continuous higher chance of acquiring a lethal mutation, will speed up the elimination of the high mutation rate genotype (IM2) and enhance the fixation rate of the low mutation rate genotype (IM3).

To confirm the role of cellular mortality as modulator of mutation rates, we selected an evolved intermediate point of line E9, with a high mutation rate resulting from fixed MMR mutations (*Figure 2—figure supplement 3*). Next, we re-evolved this population for 150 generations on the same percentage of ethanol, without further increasing this concentration when adaptation occurs, to mimic and prolong plateau conditions experienced in the original evolution experiment. We observed a fast decrease in population mutation rate (*Figure 7a*). Live-dead staining on both initial and endpoints, shows a decrease in cellular mortality linked to the decrease in mutation rate (*Figure 7b*). Further, we increased the mutation rate again by deleting the *mutS* gene (*Figure 7c*). By monitoring the number of viable cells, we observed a much higher death rate for the strain with increased mutation rate (END Δ*mutS*) (*Figure 7d*). Moreover, the strain with a low mutation rate rapidly outcompetes the Δ*mutS* strain in direct competition under 7% EtOH (*Figure 7—figure supplement 1*). Again, we used the relative fitness to calculate the theoretical number of selection rounds needed for the END strain to outcompete the END Δ*mutS* strain (*Figure 7—figure supplement 2*). In contrast to *Figure 6—figure supplement 3* the calculated rounds now correspond to the actual observed rounds. Here, both strains are isogenic apart from the *mutS* deletion, so the fitness only reflects the benefit of the anti-mutator (an intact *mutS* gene) that leads to a lower mutation rate and lower mortality. In summary, these results show that mutation rate and mortality are crucial factors to explain the fast increase of genotypes with a low mutation rate and mortality when the strain is already adapted to the environment.

## Discussion

By using experimental evolution, we observed rapid emergence of hypermutation during de novo adaptation to near-lethal ethanol stress. While mutators only sporadically occur in laboratory evolution experiments using mild stress conditions (*Sniegowski et al., 1997*; *Barrick et al., 2009*; *Sandberg et al., 2014*), all highly ethanol-tolerant lines in our study acquired a hypermutation phenotype. We provide evidence that lethal environments trigger a shift in the optimal balance between keeping a constant genetic load and mutational supply towards a higher supply rate. Despite the burden of additional, possibly lethal mutations, the increased mutational supply enables fast adaptation of at least some individuals and rescues the population from extinction (*Bell and Gonzalez, 2011*; *Lindsey et al., 2013*; *Gonzalez et al., 2013*; *Gonzalez and Bell, 2013*). Unexpectedly, by measuring the mutation rate at different time points during evolution, we found a highly dynamic

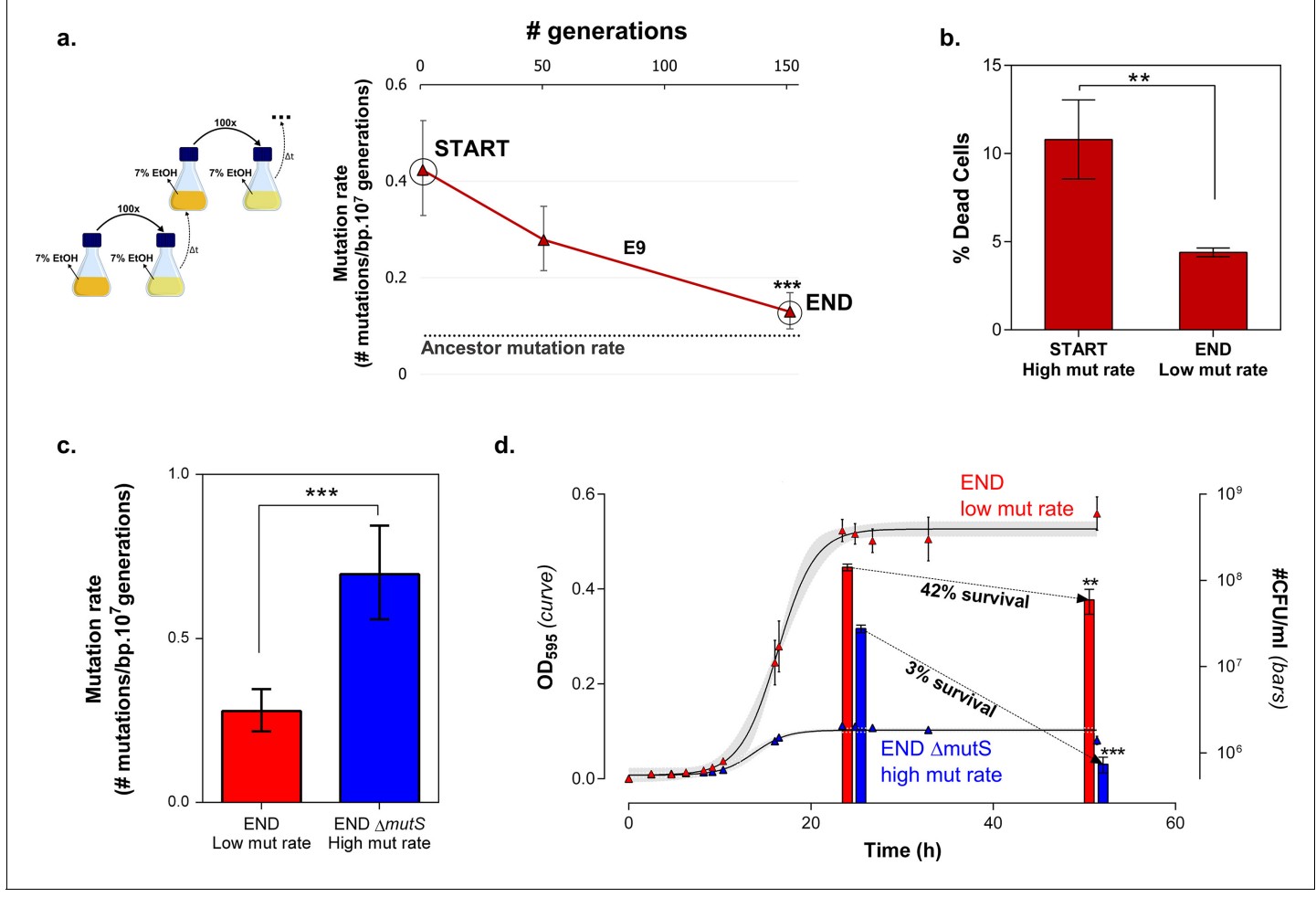

**Figure 7.** Mortality is the cost of hypermutation when adaptation to a certain EtOH stress level is achieved. (a) A selected intermediate point of evolved highly EtOH-tolerant line E9 with an increased mutation rate was evolved during approximately 150 generations on the same percentage of 7% EtOH. After 150 generations the genomic mutation rate, measured in the absence of ethanol, significantly decreased to almost the ancestral mutation rate (mean ± 95% c.i., ***p<0.001, see Materials and methods). (b) Measuring the percentage of dead cells revealed a higher death rate in the START point with high mutation rate compared to the END point with a lower mutation rate (two-sided Student's t-test, mean ± s.d., n = 3, **p<0.01). (c) To confirm the role of mortality as modulator of cellular mutation rate, the *mutS* gene was deleted in the END point with a low mutation rate. This deletion caused a significant increase in mutation rate (mean ± 95% c.i., ***p<0.001, see Materials and methods). However, the increase in mutation rate is less pronounced as for the *mutS* deletion mutant in the clean wild-type background, suggesting the presence of mutations that not only compensate for the original mutator mutation in E9 but also for a deletion of *mutS*. (d) The number of viable cells decreases significantly for both low and high mutation rate variants during growth on 7% EtOH, although this decrease in the strain with a low mutation rate is less compared to the strain with a high mutation rate (two-sided Student's t-test, mean ± s.d., n = 3, **p<0.01, ***p<0.001). These results show a lower mortality for a strain with a lower mutation rate, resulting in a competitive advantage in an EtOH environment to which the strain is already adapted.

The following figure supplements are available for figure 7:

**Figure supplement 1.** In direct competition under 7% EtOH the strain with a low mutation rate outcompetes the hypermutating strain.

**Figure supplement 2.** The theoretical number of selection rounds needed for END *ΔmutS* to fix corresponds to the actual observed number of selection rounds.

---

mutation rate that recurrently increases as a response to enhanced ethanol pressure and decreases again once cells are adapted to the stress.

The rise of hypermutation during adaptation to near-lethal ethanol stress is possibly linked to the idea of second-order selection as suggested by the growth rate and lag time measured for a

collection of mutator mutants under 5% ethanol stress (*Figure 1*; *Figure 1—figure supplement 3*). In addition, it has been reported previously that random occurrence of mutator mutations in the population is facilitated by a wider and less deleterious-mutation-biased distribution of fitness effects in changing environments (*Hietpas et al., 2013*). As a consequence of the lowered relative mutational load, populations in harsh environments may thus consist of cells with various mutation rates as they tolerate more hypermutators, possibly offering a valid additional explanation for the increased mutation rate observed in the ethanol tolerant lines (*Figure 3*). However, we observed that mutator mutations are fixed in the end point populations (*Figure 2—figure supplement 3*) and that non-mutator lines are not able to adapt to high tolerance levels (*Figure 3*), rather pointing to the former hypothesis of second-order selection or even a combination of the two explanations. In this scenario, the initial emergence of mutators is facilitated in populations exposed to severe stress (*Hietpas et al., 2013*) followed by hitchhiking of the mutator mutation along a physically linked (combination of) beneficial mutation(s) where selection act on (*Woods et al., 2011*). Additionally, this would mean that mutator mutations do not have a direct selective advantage themselves, but instead are only beneficial through enabling rapid adaptation by increasing the mutational supply rate. However, the following elements in our results might also suggest potential direct effects of the mutator mutations. First, mainly point mutations were identified in the mismatch repair genes (*Figure 2—figure supplement 3*) although inactivation of a gene is more likely to occur, given the high competitive benefit of the $\Delta mutS$ strain compared to the $mutS_{G100A}$ strain in the presence of 5% ethanol (*Figure 4—figure supplement 1*). This would suggest selection of specific changes in the mechanism of the mismatch repair pathway. Second, both the $\Delta mutS$ and the $mutS_{G100A}$ strains still increase in frequency when competed against the wild type in a ratio of 1:1000 or lower (*Figure 4*). Given the 10- to 50-fold increased mutation rate, a mutator subpopulation at a ratio of 1:1000 or lower should be too small to have an increased chance of acquiring a beneficial mutation compared to the wild-type subpopulation. These data suggest direct beneficial effects of MMR mutations (*Cooper et al., 2012*; *Torres-Barceló et al., 2013*; *Nowosielska and Marinus, 2008*) that, combined with second-order selection, can explain our observations. In addition, we confirmed that any increase in mutation rate, irrespective of the disrupted cellular system, can confer a selective benefit. Therefore, these direct effects, which are usually the result of disruptions of one specific system or even of one specific gene, may influence which specific mutator mutations eventually spread, but will only have a limited effect on the initial selection of hypermutation compared to the direct effect of linked beneficial mutations. However, at later stages these direct effects possibly affect the fate of hypermutators by lowering the cost of the extended buildup of genetic load.

Surprisingly, in addition to the increase in mutation rate, we found that hypermutator states are transitory and mutation rates decrease again once cells are adapted to the stressful environment. Current knowledge of the mechanisms underlying these changes in mutation rate remains largely fragmentary (*Raynes and Sniegowski, 2014*). In this study, we identified cellular mortality as major modulator of the population mutation rate. A higher mutation rate is linked to a higher mortality, probably due to the extended buildup of genetic load and increased probability of acquiring a lethal mutation. However, mutation rate is not the only factor affecting the mortality. Some direct effects of already accumulated mutations might explain the inconsistency between the 2-fold difference in mortality between IM2 and IM3 with a 10-fold difference in mutation rate (*Figure 6*) and the more than 10-fold difference in mortality between END and END $\Delta mutS$ with only a 2.5-fold difference in mutation rate (*Figure 7*). Notwithstanding this non-linearity, differences in mortality confer a selectable pressure that favors strains with lower mutation rates when cells are adapted to the environment. Selection of lower mutation rate genotypes that arise in an adapted population of high mutation rate genotypes is probably enhanced by the faster decrease of the high mutation rate genotypes compared to the low mutation rate genotypes due to their differences in mortality. Therefore, these findings might explain the recurrent mutation rate alterations observed in our evolution experiment. Nevertheless, the speed of mutation rate alterations clearly differs from an earlier report, showing that a single gradual decrease in mutation rate, due to invasion of a *mutY* anti-mutator in a *mutT* mutator line, occurred over a relatively long time span of at least 1000 generations (*Wielgoss et al., 2013*). While it was difficult to observe fitness benefits of anti-mutators under these less restrictive stress conditions, we here report the observation of much higher benefits under near-lethal ethanol conditions, allowing rapid, mortality-driven changes of the mutation rate.

Finally, by analyzing growth characteristics of a panel of mutators, we observed a range of mutation rates enabling fast growth under near-lethal ethanol stress. These results substantiate the theoretical modelling work of Bjedov *et al.* that predicts the highest fixation probability for a 10- to 100-fold increased mutation rate and decreasing fixation probabilities for weaker or stronger mutators (*Bjedov et al., 2003*) (*Figure 1—figure supplement 3*). Furthermore, our work extends the findings by Loh *et al.* showing that different PolA mutants with altered mutation rates predominate after serial passage in a fluctuating environment (*Loh et al., 2010*). In contrast to this study, we used genes of several distinct cellular pathways to enhance mutagenesis. That way, we were able to demonstrate that even minor alterations in mutation rate, irrespective of the targeted cellular system, can confer a competitive advantage under near-lethal, complex stress.

Both these observations corroborate the idea that moderate mutators will be more easily selected for, because their benefit is higher than low mutation rate variants and their long-term cost is lower than high mutation rate variants. The identification of mostly point mutations leading to amino acid changes and not to nonsense mutations in the MMR genes during evolution similarly suggests selection for mild increases in the mutation rate (such as shown for the $mutS_{G100A}$ mutant).

Interestingly, the occurrence of hypermutation under extreme stress is not only limited to prokaryotes, such as *E. coli*. Previously, hypermutators were also observed in *S. cerevisiae* during evolution under ethanol stress (*Voordeckers et al., 2015*), in the malaria-causing parasite *Plasmodium falciparum* (*Lee and Fidock, 2016*; *Gupta et al., 2016*), the fungal pathogen *Candida glabrata* (*Healey et al., 2016*) and in temozolomide-treated, relapsed glioblastoma tumors (*Wang et al., 2016*). These examples demonstrate the relevance of hypermutation in eukaryotes exposed to severe stress. However, the lower emergence of mutators compared to our study may be explained by the ploidy of eukaryotic cells and their larger genetic arsenal (*Thompson et al., 2006*), which allows for more alternative adaptive routes to cope with stress.

Even though we mainly focused on increased mutation rates in this study, the 12 slowly-mutating, low ethanol tolerant lines might be an interesting starting point for further research. The lack of hypermutation in these lines seems to impede further adaptation to high ethanol concentrations. Sequence analysis of two of these lines revealed the presence of mutations in *rpoZ* (subunit of the RNA polymerase) and *infB* (protein chain initiation factor) in lines E4 and E17, respectively. Since ethanol is toxic through its effect on transcription and translation (*Haft et al., 2014*), disruption of the transcription or translation machinery due to these mutations might cause increased sensitivity to higher ethanol levels. This would prevent further growth and the possibility of acquiring a mutator allele or any other mutation. Although we have no evidence supporting that the *rpoZ* and *infB* mutations are causal for the decreased mutation rate, these mutations are interesting and might explain the lack in further adaptive improvement in lines E4 and E17 (*Figure 3a*).

In conclusion, while an organism's mutation rate is generally considered a slowly-evolving parameter, we demonstrate an unexpected flexibility in cellular mutation rates matching changes in selective pressure to avoid extinction under near-lethal stress. Bacteria undergoing antibiotic treatment or cancer cells exposed to chemotherapy are prime examples of cells exposed to stressful conditions. Therefore, hypermutation should be considered a risk for both the development of multidrug resistance in pathogenic bacteria (*Hammerstrom et al., 2015*; *Jolivet-Gougeon et al., 2011*; *Blázquez, 2003*; *Chopra et al., 2003*) and cancer relapses as recently shown (*Wang et al., 2016*). Targeting hypermutation could pave the way not only for the development of novel anti-cancer therapies, but also for containing the spread of multidrug tolerant pathogens and even for the generation of robust, stress-resistant strains for use in various industrial processes.

## Materials and methods

### Bacterial strains and culture conditions

*E. coli* SX4, SX25, SX43 and SX43 Δ*venus*, all derived from BW25993 (*Datsenko and Wanner, 2000*), were used in this study. SX4 is characterized by a genomic *tsr-venus* fusion inserted in the *lacZ* gene under control of the *lac*-promoter (*Yu et al., 2006*). SX43 is a derivative of SX4 in which the kanamycin-resistance cassette (Km$^R$) was removed by Flp-mediated recombination (*Van den Bergh et al., 2016*; *Cherepanov and Wackernagel, 1995*). The *tsr-venus* fusion results in a polarly localized fluorescent Venus tag. *E. coli* SX25 expresses a genomic *venus* marker inserted in the *lacZ* gene under

control of the *lac*-promoter (*Yu et al., 2006*). SX43 Δ*venus* is a non-fluorescent variant constructed by P1*vir* transduction (*Thierauf et al., 2009*) using the *lacI::KmR* Keio mutant (Keio collection number JW0336) (*Van den Bergh et al., 2016*; *Baba et al., 2006*) as donor. All strains were grown in an orbital shaker at 200 rpm and 37°C in liquid lysogeny broth (LB) medium or on LB agar plates.

## Construction of deletion mutants

Target genes to increase the genomic mutation rate were selected based on their various roles in DNA replication and repair (*Supplementary file 1A*). Hypermutating variants of the ancestor were generated by P1*vir* transduction (*Thierauf et al., 2009*) to the SX43 ancestor using the corresponding Keio deletion mutants as donor strains (*Baba et al., 2006*) (RRID:SCR_002303). Transductants were subsequently selected on kanamycin resistance. Correct deletion of the target genes in positive colonies was confirmed by PCR (*Supplementary file 1B*). The *mutS* deletion was introduced using the protocol described by Datsenko and Wanner (*Datsenko and Wanner, 2000*). In short, a kanamycin-resistance cassette flanked by FRT sites was amplified from the pKD4 plasmid using primers with homologous ends complementary to the flanking sequences of the *mutS* gene (*Supplementary file 1B*). This PCR product was electroporated in the ancestor in which the λ-red genes for homologous recombination were expressed from the pKD46 plasmid. Positive colonies were selected on kanamycin resistance and correct deletion of the *mutS* gene was assessed by PCR.

## Determination of near-lethal EtOH percentage

To assess the percentage of EtOH needed to expose cells to near-lethal stress, the growth dynamics of wild-type *E. coli* in different levels (0–5% (v/v)) of EtOH were studied. Data were fitted to a Gompertz equation for bacterial growth dynamics (see *Equation 1*) to extract relevant growth parameters (*Figure 1—figure supplement 2*). 5% EtOH was considered a breakpoint concentration as an abruptly increased doubling time and decreased carrying capacity is observed under these conditions. To ensure the proper percentages of ethanol added to the medium we used the Alcolyser beer analyzing system (Anton Paar GmbH, Austria).

## Determination of lag time and growth rate

Lag times and growth rates of selected mutants were determined using the Bioscreen C system for automated monitoring of microbial growth (Bioscreen C MBR, Oy Growth Curves AB Ltd., Finland) (RRID:SCR_007172). Cells were grown in $10 \times 10$ well Honeycomb microplates with shaking at 37°C and optical density at $A_{600nm}$ was automatically measured every 15 min. Growth in each well was monitored for five days. First, the optical density of each preculture was equalized to approximate equal numbers of cells for all mutants. Next, dilution series were made ranging from $10^{-1}$ to $10^{-4}$. To test the effect of different starting amounts of cells on the lag time in the presence of EtOH, 20 μl of each dilution was used to inoculate 180 μl of LB medium supplemented with 5% EtOH. These dilutions correspond to the different inoculum sizes as shown in *Figure 1* and *Figure 1—figure supplement 3*. For every mutant all dilutions were tested in biological triplicate. Additionally, each well was covered with 100 μl of mineral oil (BioReagent, Sigma-Aldrich, MO, USA) to prevent evaporation of EtOH as previously described (*Zaslaver et al., 2006*). As previously shown (*Kaplan et al., 2008*; *Zaslaver et al., 2004*; *Ronen et al., 2002*) mineral oil has no significant effect on growth or aeration. Growth curves were fitted using the widely accepted Gompertz equation as previously described (*Zwietering et al., 1990*). In this equation, $y_0$ is the starting density, $y_M$ is the carrying capacity of the population, SGR is the specific growth rate ($h^{-1}$) and LT is the lag time (h). $Log_{10}$ of the optical density values (595 nm) were used to fit with this equation and subsequently extract all growth parameters.

$$y(x) = y_0 + y_M * e^{\left[-e^{\left(\left(2.718 * \frac{SGR}{yM}\right)*(LT-x)+1\right)}\right]}$$

(1)

Equation 1: Gompertz equation for fitting of bacterial growth dynamics.

## Competition assay

To determine the relative fitness of a mutant compared to the wild type, we conducted direct competition experiments. All mutator mutants carried the Venus fusion that could be detected by

excitation at 530 nm, while the ancestor SX43 Δ*venus* was not fluorescent. The relative fitness of all mutants was assayed in triplicate. Both non-fluorescent ancestor and fluorescent mutator mutants were revived from a glycerol stock stored at −80°C and grown overnight at 200 rpm and 37°C. These overnight cultures were first diluted to an $A_{595nm}$ of 0.5 to obtain an equal cell quantity for all cultures prior to the competition experiment. Next, equal amounts of ancestor and mutant were diluted 1000x in 50 ml LB medium, containing 5% EtOH. To avoid EtOH evaporation, cultures were grown in flaks with a rubber-sealed screw cap. To start the competition experiment with different mutator versus ancestor ratios, corresponding amounts of both strains were mixed. The initial ratios were verified by flow cytometry using a BD Influx cell sorter equipped with a 488 nm laser (200 mW) and standard filter sets (530/40 nm for Venus detection). To assess relative fitness, mixed populations were incubated for 48 hr in the presence of 5% EtOH. Ratios of both ancestor and wild type were subsequently determined by flow cytometry. For each sample at least 100.000 cells were counted and each cell was tallied as ancestor or mutator based on their fluorescence intensity. Loss of fluorescence in the mutator mutants was accounted for by measuring fluorescence and loss of fluorescence in the individually grown ancestor and all mutator mutants. First, the fraction false positive non-fluorescent mutators was quantified for each mutant. This fraction was subsequently used to correct the measured mutator versus wild type ratios to account for non-fluorescent mutators.

As the results of this test reflect ratios of ancestor and mutant in the population before and after growth on 5% EtOH, the proportion of a less fit population is expected to decline relative to the other. Therefore, we calculated the relative fitness, W, as previously described (*Equation 2*) (*Van den Bergh et al., 2016*). This equation is based on a discrete time-recurrence equation that describes the spread of mutant in a haploid population thereby defining relative fitness based on differences in survival between two strains (*Van den Bergh et al., 2016*; *Otto and Day, 2007*). This equation allows to calculate the relative fitness, $W_A$, of a mutant A compared to its competitor, based on the difference between the final detected proportion ($A_{end}$) and the initial proportion ($A_{start}$) of the mutant in the population, given a certain number of selection rounds n. Significance of difference from 1, where the mutant has no benefit over the wild type, was determined using a repeated measures ANOVA with post hoc Dunnett correction. An F-test was used to assess the difference in variance between the groups that were statistically compared. In order to confirm marker neutrality, we competed the fluorescent ancestor (SX43) and non-fluorescent ancestor (SX43 Δ*venus*) against each other both in the absence and presence of EtOH. No significant difference from one was observed using a one-sample Student's t-test (data not shown). Additionally, the neutrality of the Venus marker was already confirmed in a previous study (*Van den Bergh et al., 2016*).

$$W_A = \frac{1}{\left[ \frac{\left( (1-A_{end}) \times A_{start} \right)}{(A_{end} \times (1-A_{start}))} \right]^{\left(\frac{1}{n}\right)}}$$

(2)

Equation 2: Relative fitness of mutant A compared to its competitor

## Experimental evolution

The 20 parallel populations of the evolution experiment originated from independent colonies of the ancestral strains SX4 or SX25. Odd lines were founded by SX4, while even lines were founded by SX25 to enable detection of cross-contamination between parallel lines. All strains were initially grown in LB medium containing 5% EtOH. This percentage was found to mimic near-lethal stress (*Figure 1—figure supplement 2*). The culture volume during the evolution experiment was 50 ml and dilution was 100-fold at each transfer. Ethanol tolerance of a population was measured as the ability to grow in liquid medium in the presence of a certain percentage of ethanol to an optical density of at least 0.2. Intermediate time points, sampled every transfer to fresh medium, were stored in a −80° glycerol stock. Growth in exponential phase was maintained throughout the evolution experiment to select for growth rate and lag time only and to minimize potential, unwanted effects and genomic changes due to stress that would additionally be experienced by nutrient limitation. Two different parameters were used to monitor the evolution of the independent lines. First, the optical density was measured. A strain with an $A_{595nm}$ around 0.2 was assumed to be in exponential phase. Second, the time needed for the strain to achieve exponential growth at an $A_{595nm}$ around 0.2 was used to determine the degree of adaptation. When the population reached exponential growth within 24 hr, we assumed it was fully adapted to a certain percentage of EtOH. We transferred the

population to new LB medium containing 0.5% more EtOH than in the previous step. By increasing the percentage of EtOH during the adaptation of the populations, the near-lethal stress was maintained. In later stages of the evolution experiment at very high EtOH concentrations of 7.5% or more, an increase of 0.5% was found to be excessive. Therefore, we increased the percentage with 0.25% EtOH starting from 7.5% EtOH tolerance. If the strain needed more than 24 hr but less than 14 days to reach exponential growth, we assumed that it was not yet completely adapted to a certain concentration of EtOH. Therefore, we transferred the strain to new LB medium containing the same percentage of EtOH as in the previous step. If the strain needed more than 14 days to grow, we assumed this population died out. Therefore, we revived the previously stored time point and used it to restart the evolving line in new LB medium with 0.5% EtOH less than the tolerance level of this time point (*Figure 2—figure supplement 1*). The minimal optical density of 0.2 upon transfer resulted in an average final cell density of $5.4 \times 10^8$ CFU/ml in a volume of 50 ml. For each passage, we consequently transferred 500 µl or approximately $2.73 \times 10^8$ CFUs ($N_0$). The average number of generations ($g$) for each growth cycle, estimated based on the optical density reached upon transfer, is 6.67. Taking these values together, we can calculate an estimated effective population size ($N_e$) of $1.82 \times 10^9$ using the formula (*Lenski et al., 1991*[91]) $N_e = gN_0$. Finally, the number of generations are estimates calculated with a previously described equation, assuming equal growth of the entire population based on optical density and time (*Wiser et al., 2013*) (*Equation 3*). In this equation $CFU_i$ is the number of viable cells at the start of each cycle, while $CFU_e$ is the number of viable cells at the end of each cycle and c is the total number of cycles. The number of viable cells was estimated using optical density ($A_{595nm}$) values. This calculation does not specifically take into account the death rate of the cells. However, since the optical density reflects both living and death cells in the culture, calculating the number of generations using the OD values is more accurate compared to calculations based on CFU counts. Indeed, when using the viable cell count data of IM1, IM2 and IM3 (*Figure 5*), we found a 1.27-fold ($\pm 0.29$) underestimation of the number of generations when using the number of viable cells as compared to using OD values.

$$\sum_{c=1}^{n} log_2 \left( \frac{CFU_e}{CFU_i} \right) \tag{3}$$

Equation 3: Estimation of the number of generations per cycle based on the initial and final number of viable cells.

## Fluctuation assay

The genomic mutation rate of strains of interest was estimated using a Luria-Delbrück fluctuation assay. This assay is commonly used to determine the spontaneous mutation rate at different loci in the genome, where mutations cause easily-scored phenotypic changes. Acquiring rifampicin resistance through mutations in the *rpoB* gene was used as a measurable marker. This protocol was adapted from the one described by Jeffrey Barrick (http://barricklab.org/twiki/bin/view/Lab/ProtocolsFluctuationTests). The selected strains were revived from a glycerol stock stored at −80°C and grown overnight in an orbital shaker at 200 rpm and 37°C. All strains were tested in at least two independent biological replicates. The number of cells in each culture was determined using an optical density versus cell count standard curve and was subsequently equalized over all tested strains. Next, the equalized cultures were diluted 100 times in fresh LB medium and grown in an orbital shaker at 200 rpm and 37°C for 2–3 hr until the optical density at 595 nm reached 0.2–0.4. At this optical density the final cell density in solution did not exceed $2-4 \times 10^8$ CFU/ml. These preconditioned strains were then diluted in LB medium to a density of 5000 cells per ml, which is denoted as the master inoculum mix. The master inoculum mix was divided in replicate cultures of 200 µl in separate Eppendorf tubes or a 96-well plate. For each strain at least 30 replicate parallel cultures, divided over minimum two biological repeats were used to determine the number of spontaneous mutants. These cultures were grown for 24 hr and plated on LB agar supplemented with 100 µg/ml rifampicin to determine the number of spontaneous mutants that arose during the growth period. Additionally, for each biological repeat, at least four cultures were grown for 24 hr, diluted and plated on LB agar to determine the total number of viable colonies. The colonies on the non-selective LB agar plates were counted after 24 hr, while colonies on the selective, rifampicin plates were first counted after 48 hr and again after 72 hr. The number of mutants divided by the total number

of cells gives a mutation rate estimate of the tested strain. The occurrence of rifampicin resistance conferring mutations in the *rpoB* is extrapolated to estimate the global genomic mutation rate. In recent years, many improvements were made to the statistical estimation of mutation rates based on the number of mutants and the total number of cells. We used the Ma-Sandri-Sarkar Maximum Likelihood Estimation method as implemented in the Fluctuation Analysis Calculator (FALCOR, http://www.keshavsingh.org/protocols/FALCOR.html) (*Hall et al., 2009*). For the statistical analysis on the mutation rate estimates, 95% confidence intervals, calculated by the FALCOR, were compared. In the case of confidence interval overlap, mutation rates were statistically compared using a two-sided Student's t-test on the normally distributed absolute number of mutational event as calculated by FALCOR (*Hall et al., 2009*). To ensure correct comparison of the mutation rates, we verified that the population densities at the time of plating on rifampicin did not differ significantly. If this was not the case, the test was repeated. The statistical difference between the population densities was measured using a one-way ANOVA with post-hoc Tukey correction. We found no significant difference for any intermediate point compared to the average density and to the density of the other time points. We therefore avoid possible population density effects (*Krašovec et al., 2014*).

To study the change of mutation rates during adaptive evolution, we performed a correlation analysis between the difference in mutation rate and the difference in EtOH tolerance at each time point. Since EtOH tolerance during evolution only increases by 0.25% or 0.5%, we considered the difference in EtOH tolerance as a discrete ordinal variable. Therefore, we used the non-parametric Spearman method to determine the significance of the correlation between the difference in EtOH and the difference in mutation rate between consecutive time-points (Graphpad Prism 6, CA, USA).

## Whole-genome sequencing and identification of mutations

High-quality genomic DNA was isolated from overnight cultures of the ancestor and end points of evolved lines (DNeasy Blood and Tissue kit, Qiagen). We isolated genomic DNA from both mixed pools and one characterized clone of each high ethanol tolerant line and two low ethanol tolerant lines. Both *Figure 2—figure supplement 2* and *Figure 2—figure supplement 3* represent the results from the analysis of the pooled sequence data. Concentration and purity of the DNA were determined using Nanodrop analysis (Thermo Fisher Scientific, MA, USA), gel electrophoresis and Qubit analysis (Thermo Fisher Scientific). Libraries were prepared at GeneCore (EMBL, Heidelberg, Germany) (RRID:SCR_004473) using the NEBNext kit with an average insert size of 200 bp. The DNA libraries were multiplexed and subjected to 100-cycle paired-end massive parallel sequencing with the Illumina HiSeq2000 (RRID:SCR_010233) (GeneCore, EMBL, Heidelberg, Germany). CLC Genomics Workbench version 7.6 (RRID:SCR_011853) (https://www.qiagenbioinformatics.com) was used for analysis of the sequences. Following quality assessment of the raw data, reads were trimmed using quality scores of the individual bases. The quality limit was set to 0.01, and the maximum allowed number of ambiguous bases was set to 2. Reads shorter than 15 bases were discarded from the set. The trimmed reads were mapped (mismatch cost = 2; insertion cost = 3; deletion cost = 3; length fraction = 0.8; similarity fraction = 0.8) to the *E. coli* MG1655 reference genome (NC_000913.1) using the CLC Assembly Cell 4.0 algorithm yielding an average coverage of approximately 150x. Finally, mutations in all samples were detected using the CLC Fixed Ploidy Variant Detector. To exclude mutations in the SX4 ancestor compared to the MG1655 reference genome, we compared the mutations of all evolved lines with the SX4 ancestor.

## Mortality assay

To assess the rate at which cells die during growth, we made growth curves using optical density measurements with concurrent viable cell determination. The ancestor and selected evolved intermediate time points were directly inoculated from a frozen glycerol stock in 50 ml LB medium containing no EtOH, 5% EtOH, or 6.5% EtOH. Each strain was tested in triplicate. All flasks were subsequently grown at 200 rpm and at 37°C. At 30 different time points during a 90 hr timespan the optical density was measured and samples were taken for CFU determination. For each sample, a dilution series was made and appropriate dilutions were plated on LB agar plates using an EddyJet2 spiral plater (IUL Instruments, Spain). Agar plates were grown 48 hr at 37°C and the CFU/ml was determined using the Flash and Go automatic colony counter (IUL Instruments). During growth, the number of CFU/ml initially increases exponentially but then flattens and decreases again. The colony

count data corresponding to the decrease in CFU/ml were fitted using an exponential decay function (*Equation 4*) in GraphPad Prism 6. In this function, k is the death rate constant. For all samples, this constant was determined. Statistical significance of the difference between the death rate constants of two consecutive evolved time points was determined using a two-tailed Student's t-test.

$$y = (y_{max} - y_{min}) * e^{-k*x} + y_{min} \qquad (4)$$

Equation 4: Exponential decay function with k the decay constant

## Live-dead staining

To measure the amount of dead cells at a certain time point in a population we used the LIVE/DEAD *Bac*Light Bacterial viability kit (Thermo Fisher Scientific). The selected strains were revived from a frozen glycerol stock and grown overnight in an orbital shaker at 200 rpm and 37°C. Overnight cultures were diluted to an $A_{595nm}$ of 0.5. Next, 1 μl propidiumiodide (20 mM, Thermo Fisher Scientific) per 1 ml diluted culture was added, vortexed to mix the propidiumiodide homogeneously and incubated in the dark at room temperature for 10 min. Propidiumiodide can only penetrate the cell when the membrane is disrupted, as is the case in dead cells, and can be detected by excitation at 620 nm. Therefore, the amount of dead cells in a population could be determined by flow cytometry. All strains were tested at least in triplicate. To measure the number of dead cells throughout the different growth phases, the selected strain was inoculated at different time points ranging from 48 hr to 10 hr prior to flow cytometry analysis. The amount of dead cells was determined as previously described. Statistical significance was determined using a two-sided Student's t-test.

## Acknowledgements

TS is a fellow of the Agency for Innovation by Science and Technology (IWT). The research was supported by the KU Leuven Research Council (PF/10/010; PF/10/07; IDO/09/010; IDO/13/008; CREA/13/019; DBOF/12/035; DBOF/14/049), Interuniversity Attraction Poles-Belgian Science Policy Office IAP-BELSPO) (IAP P7/28), ERC (241426), Human Frontier Science Program (HFSP) (RGP0050/2013), FWO (G047112N, KAN2014 1.5.222.14), Flanders Institute for Biotechnology (VIB) and the European Molecular Biology organization (EMBO). We thank S. Xie for providing the ancestor strains.

## Additional information

### Funding

| Funder | Grant reference number | Author |
|---|---|---|
| Agentschap voor Innovatie door Wetenschap en Technologie | Strategic Basic Research Fellowship,121525 | Toon Swings |
| Fonds Wetenschappelijk Onderzoek | Postdoctoral Fellowship,12O1917N | Bram Van den Bergh |
| Fonds Wetenschappelijk Onderzoek | Postdoctoral Fellowship,1249117N | Karin Voordeckers |
| Onderzoeksraad, KU Leuven | PF/10/010 | Kevin J Verstrepen Jan Michiels |
| H2020 European Research Council | 241426 | Kevin J Verstrepen |
| Human Frontier Science Program | RGP0050/2013 | Kevin J Verstrepen |
| Vlaams Instituut voor Biotechnologie | | Kevin J Verstrepen |
| European Molecular Biology Organization | | Kevin J Verstrepen |
| Onderzoeksraad, KU Leuven | IDO/09/010 | Kevin J Verstrepen Jan Michiels |

| Onderzoeksraad, KU Leuven | DBOF/12/035 | Kevin J Verstrepen<br>Jan Michiels |
|---|---|---|
| Onderzoeksraad, KU Leuven | DBOF/14/049 | Kevin J Verstrepen<br>Jan Michiels |
| Onderzoeksraad, KU Leuven | CREA/13/019 | Maarten Fauvart |
| Fonds Wetenschappelijk On-<br>derzoek | KAN2014 1.5.222.14 | Maarten Fauvart |
| Federaal Wetenschapsbeleid | Interuniversity Attraction<br>Poles Programme P7/28 | Jan Michiels |
| Fonds Wetenschappelijk On-<br>derzoek | G047112N | Jan Michiels |
| Onderzoeksraad, KU Leuven | IDO/13/008 | Jan Michiels |

The funders had no role in study design, data collection and interpretation, or the decision to submit the work for publication.

## Author contributions

TS, Conceptualization, Data curation, Formal analysis, Supervision, Validation, Investigation, Visualization, Methodology, Writing—original draft, Writing—review and editing, T.S. conceptualized the study, designed and performed the experiments, analyzed the data and wrote the manuscript; BVdB, Conceptualization, Supervision, Methodology, Project administration, Writing—review and editing, B.V.D.B. discussed the results and edited the manuscript; SW, Data curation, Investigation, S.W. helped in performing the experiments; EO, Data curation, Investigation, E.O. helped in performing the experiments; KV, Conceptualization, Funding acquisition, Project administration, Writing—review and editing, K.V. conceptualized the study, discussed the results and edited the manuscript; KJV, Conceptualization, Funding acquisition, Project administration, Writing—review and editing, K.J.V. conceptualized the study, discussed the results and edited the manuscript; MF, Conceptualization, Supervision, Funding acquisition, Project administration, Writing—review and editing, M.F. conceptualized the study, designed the experiments, discussed the results and edited the manuscript; NV, Conceptualization, Supervision, Funding acquisition, Investigation, Methodology, Project administration, Writing—review and editing, N.V. conceptualized the study, designed the experiments, discussed the results and edited the manuscript; JM, Conceptualization, Supervision, Funding acquisition, Investigation, Methodology, Project administration, Writing—review and editing, J.M. conceptualized the study, designed the experiments, discussed the results and edited the manuscript

## Author ORCIDs

Toon Swings, http://orcid.org/0000-0002-1225-3377
Natalie Verstraeten, http://orcid.org/0000-0002-9548-4647
Jan Michiels, http://orcid.org/0000-0001-5829-0897

# Additional files

## Supplementary files

• Supplementary file 1. Addtional tables with primer sequences and selected mutator genes.

## Major datasets

The following dataset was generated:

| Author(s) | Year | Dataset title | Dataset URL | Database, license, and accessibility information |
|---|---|---|---|---|
| Swings T, Michiels J | 2016 | Highly ethanol tolerant Escherichia coli with hypermutation phenotype | https://www.ncbi.nlm.nih.gov/bioproject/PRJNA344553 | Publicly available at the NCBI SRA repository (accession no: PRJNA344553) |

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
