## [Decision Letter]

Thank you for submitting your article "Adaptive tuning of mutation rates allows fast response to lethal stress in *Escherichia coli*" for consideration by *eLife*. Your article has been reviewed by three peer reviewers, one of whom Wenying Shou (Reviewer #1), is a member of our Board of Reviewing Editors, and the evaluation has been overseen by Diethard Tautz as the Senior Editor. The following individuals involved in review of your submission have agreed to reveal their identity: Olivier Tenaillon (Reviewer #2).

The reviewers have discussed the reviews with one another and the Reviewing Editor has drafted this decision to help you prepare a revised submission.

Summary:

This article shows that mutation rate can be highly dynamic during evolution. The work is interesting. However, there are major issues that will need to be clarified and addressed. Details are provided in the reviewer comments below. Please note that depending on your response, the reviewers might come to the conclusion that additional experiments could be necessary.

Reviewer #1:

This review is from the point of view of a "general audience" member who is informally interested in mutators. This paper showed that when *E. coli* was subjected to lethal ethanol stress, mutators evolved in multiple lines. Those mutator lines (but not lines with normal mutation rates) could tolerate lethal levels of ethanol. Mutation rates in these mutator lines were then reduced in later stages of evolution.

Authors claim that "an organism's mutation rate is often considered to be a slowly-evolving parameter." I am not sure whether this claim is true since Wielgoss *et al.*, 2013 showed that a hyper-mutator evolved in one of the 12 Lenski lines and that it was later replaced by milder mutators. The difference is that in this study, reduction of mutation rate is through compensatory mutations instead of replacement of one type of mutator with another type. Hammerstrom *et al.*, 2015 (not cited) showed that hypermutators can repeatedly evolve in response to tigecycline (via transposon insertion). Thus, repeated evolution of hypermutator (in contrast to the stochastic appearance of mutator in one of Lenski lines) is not new either. However, there are nice experiments in this paper, and the dynamics of mutation rate is also tracked with fine resolution.

Main points:

Subsection “Long-term adaption to high ethanol stree in *E. coli* is contingent upon hypermutation”: "evolutionary dead ends" – how do you know that? Have you introduced mutator mutations into dead-end lines and observed lack of improved ethanol survival? If so, it will be interesting to know why the "dead-end" lines failed to evolve mutator phenotypes.

Quantification method of ethanol tolerance needs to be described. Figure 5: Units? I am in fact not sure about the point of Figure 5 given that high ethanol tolerance can be achieved with low mutation rate at later stages of evolution.

I do not find increased death rate associated with mutators particularly interesting, without authors explaining to me the fundamental difference between slower birth, increased death, or a combination of the two. After all, one can view birth and death as "net-growth".

I do not understand the point of Table 1. Why not recalculate estimated number of mutations based on the dynamics of mutation rate and see whether it matches observed number of mutations?

It will be interesting to know which mutations directly promote ethanol tolerance (if you already have the data).

Reviewer #2:

In their manuscript entitled "Adaptive tuning of mutation rate allows fast response to lethal stress in *E. coli*", Toon Swings and collaborators study the evolution of mutation rate under adaptation to ethanol. They show that increased mutation rate is rapidly selected for in the early stages of the adaptation, and more surprisingly they uncover a rapid decay of mutation rate that seems to be directly linked to some increased mortality.

The data are solid, quantitative and very interesting, especially the connection between mortality and mutation rates. My comments are mostly on some of the interpretations of the results, and on some experimental details

In the Introduction, it is not fully clear if ethanol 5% is a real lethal stress: mutations are required to survive the first growth cycle. Based on the lag time is seems so. If it is that case the paper would benefit from referring to the filed of rescue experiments. These are fashionable experiments that study adaptation in conditions in which it is essential for survival. A paragraph (or even the title) on the link between the experiment and that field would increase the readership of the paper.

The paper is clearly showing a new factor that may explain changes in mutation rate: mortality. However, the whole interpretation of the phenomenon is still linked to the idea of second order selection and load. This is really in line with the first theoretical data on mutators and second order selection that neglected all potential direct effects. More recent work have put more emphasis on potential direct effects and I think the discussion should introduce some of these ideas and eventually challenge them. Some results here suggest that eventually some direct effect of mutators may be at play, not in the early selection of mutators but eventually in their costs.

First in Figure 4 even at extremely low frequencies, mutator alleles are selected for. This is quite surprising as at some point mutator subpopulation should be to low to generate any beneficial mutations. Indeed if we assume a 1000-fold effect on transitions, mutator should be outcompeted by wild type at a ratio of 1/10000 or so. So Figure 4 suggest a direct effect. Also mentioned in the text the fact that only point mutations were found in *mutS* and no inactivation is also an argument for a more complex effect than just increase in mutation rate. (inactivation should be much more likely to occur (though some deletions may not be detected at population level with the pipeline used) and as they have an impressive benefit in competition they should invade immediately…

Even in the very nice experiments linking mortality and mutation rate, a 10 fold reduction in mutation rate from IM2 to IM3 lead to a two fold decrease in mortality or less in dead cells. So mutation rate is not the only player. Mutation could have accumulated that lead to the mild differences between IM2 and IM3, But in Figure 7, the author found that a three fold change in mutation rate has this time a drastic effect on survival (more than 10 times)And the growth curves are completely different.

It may be extremely difficult to go after mechanistic effects, but discussing that both primary selection and second order selection may be at play will benefit the paper.

MMR have been shown to be beneficial because they allow a fast switching of flagella expression though increased recombination, have been found to be toxic when too many mismatches are present in the cell such that they lead to double strand break… and few other recent papers have suggested that evolution of mutation rate may also be linked to direct effect. This does not decrease in any way the value of the experiments done, but is worth mentioning in the discussion.

Of note, with respect to potential rise of mutators, the experiments with a subset of mutators involved in different mechanisms is clearly a proof that increased mutation rate is selected for. Then the fact that only MMR clones are found and no KO is found suggests that there are eventually selection of more moderate effects or of some specific mecanisms.

Overall it is a very nice study that clearly shows a very dynamic and fast evolution of mutation rate that is extremely relevant to understand evolution in stressful conditions.

Reviewer #3:

The authors present an interesting set of experiments showing surprisingly high dynamics of changes in the mutation rate of *E. coli* populations adapting to near-lethal ethanol concentrations. Several previous microbial evolution experiments have demonstrated the rapid rise of mutator mutants via genetic linkage with beneficial mutations during adaptation to novel conditions and stresses. Some studies also found declines of the mutation rate later during adaptation, when the ratio of beneficial to deleterious mutations declines, due to the selection of anti-mutator genotypes. However, the selection of genotypes with lower mutation rates is thought to be driven by much smaller selection coefficients than the selection of genotypes with increased mutation rates. Mutators hitchhike with beneficial mutations, which can have large fitness benefits under stressful conditions, while the fitness benefit of anti-mutators is proportional to the difference in mutation load between mutator and anti-mutator – with the difference in the fraction of lethal mutations as an upper limit. Given a wild-type mutation in of ~0.003 per genome per generation (e.g. Drake *et al.*, 1998), *mutS* mutators have a mutation rate of ~0.1 (10-100 fold higher). This sets the upper limit to the selective benefit of a change back from mutator to wild-type mutation rates to 10% if all mutations were lethal, suggesting that rapid declines of mutation rates are theoretically possible under conditions where lethal mutations are frequent. While it has been very difficult to detect fitness benefits of anti-mutators under less stressful conditions (e.g. Wielgoss *et al.*, 2013), the authors of the present study claim to have observed much higher benefits under near-lethal conditions, allowing not only rapid increases, but also subsequent rapid decreases of the mutation rate. If true (but see below), this finding is sufficiently interesting and novel to publish in a general high-impact journal, such as *eLife*.

We have three main comments:

Essential information is missing at several places, which prevents both a basic understanding of the population dynamics during the experiment and a comparison of the present findings with those of previous work on the evolution of mutation rates. First, no information on the effective population size of the evolving populations is given: what culture volumes, which dilutions, what final cell densities were obtained and what was the effective population size (~inoculum size x number of generations per growth cycle)? Second, given the high mortality of cells, how were numbers of generations calculated? Dead cells have also been produced and they should be taken into account for the estimates of the total numbers of generations. Third and perhaps most importantly, it was unclear whether fluctuation tests and genome sequence analyses have been done on individual clones or mixed population samples. As a result, we don't know how to interpret Figure 5 and Table 1: do mutation rates in Figure 5 reflect mutation rates of fixed genotypes or are these a mix of mutator and wild-type mutation rates of a polymorphic population? Similar for the frequency of observed mutations in Table 1: are these based on sequences of random clones picked form these populations or on mixed population DNA? The variation in mutation rate shown in Figure 5 suggests that these reflect population mutation rates, since changes are too small to reflect pure genotype effects if these involve mutator mutants such as mutS (see Figure 4). This distinction is crucial for the possible mechanisms that should be considered for explaining the dynamics in mutation rate: if high mutation-rate genotypes fix in the population, later observed low mutation-rate genotypes must derive from the former genotypes, whereas if they are *not* fixed, later observed low mutation-rate genotypes may derive from ancestral genotypes that were still present in the population when the increase in mutation rate was measured.

Our second comment is about the surprising rapid selection of genotypes with lower mutation rates at several times during evolution (Figure 5). As sketched above, to understand whether natural selection has driven these changes (and assuming that they reflect the mutation rate of the population, not of individual clones!), two pieces of information are required. First, the fitness consequences of these changes should be quantified in direct competition experiments between genotypes with high and low mutation rate under the conditions prevalent during evolution. Now only the production of dead cells is compared (Figure 6), but no attempt is made to translate mortality estimates into fitness values, either empirically (by running these competitions) or theoretically (by using growth models to predict these effects). The decline of mutation rate during 150 generations of continues evolution for population E9 (Figure 7), and its correlation with lower production of dead cells, are suggestive, but do not solve the riddle how mortality rates translate into fitness under the selective conditions. Second, fitness estimates of high and low mutation-rate genotypes should then be used to predict the dynamic of decline of the mutation rate, to see if this mechanism may indeed explain the dynamics shown in Figure 5.

Our third comment is on Figure 1. First, why would small inocula lead to much higher yields than large inocula? This is surprising, but remains unexplained. Second, the identity of the growth curves should be better explained. What do the grey lines reflect, individual replicates or mean curves for each mutator strain? The number of lines suggest the former, but the text refers to "the" growth curve for *dnaQ*, suggesting they reflect mean curves. It may be best to show mean growth curves or fitted growth models per strain, with different colors per strain. How are the different populations (of different size) generated? Do they come from the same 'batch' which are either less or more diluted? Or does the 10^7^ dilution come from a later time point of the smaller dilutions? The physiological state of the cells (*e.g.* early exponential phase, later exponential phase) at the moment that they are inoculated is known to affect the subsequently generated growth curves. Or perhaps, can larger population sizes 'soak' the ethanol, and thus effectively decrease the concentration in the environment? (see *e.g.* doi: 10.1098/rsbl.2012.0569)

[Editors' note: further revisions were requested prior to acceptance, as described below.]

Thank you for resubmitting your work entitled "Adaptive tuning of mutation rates allows fast response to lethal stress in *Escherichia coli*" for further consideration at *eLife*. Your revised article has been favorably evaluated by Diethard Tautz (Senior editor). The Reviewing editor Wenying Shou and one of the original reviewers (Olivier Tenaillon) are satisfied with your responses to their critiques. The original third reviewer was not able to re-review. Instead, two scientists familiar with mutation rates co-reviewed your work and your response.

The manuscript has been improved but there are some remaining issues that need to be addressed before acceptance, as outlined below:

Reviewers #4 and 5:

This is a study with complicated design and large amount of work. The authors have addressed (or at least tried) most concerns from the last round of revisions. However, as new reviewers, we have some additional points that need the authors' explanations. The paper could be accepted given satisfying responses.

1) "A higher mutation rate is linked to a higher mortality, probably due to the extended buildup of genetic load and increased probability of acquiring a lethal mutation". There could be a more straight-forward explanation: Krasovec *et al.*, (2014 Nature Communications) showed that mutation rate from fluctuation assay inversely correlates with population density, due to cell-cell interactions. For the same experimental evolution line at different intermediate time points, higher mortality could reduce population density-given the same inoculum size for fluctuation assay, which in turn increases mutation rate. This alternative interpretation needs to be addressed in the current experimental system.

2) While the authors interpret the data as evidence to support the idea that cells evolve hypermutation to avoid extinction under near-lethal stress, we should also consider a more straight-forward alternative.

In the population, mutants with various mutation rates are generated continuously. When the environment turns stressful, the distribution of fitness effects of mutations accordingly becomes wider and less deleterious-mutation-biased due to the decreased population-average fitness (*e.g.* Hietpas *et al.*, 2013). As a result, the mutation load of hypermutators relative to WT becomes smaller and hypermutators are more likely to survive or even prosper (if hitchhiked with beneficial mutations). The argument can be reversed when organisms become more adapted in the new environment to explain the decline of hypermutators. In this way, the dynamic pattern of mutation rates can be understood as a passive balance between influx (by random mutations) and outflux (due to accumulated genetic load) of mutators without evoking any active response. It would help if the authors can clarify if and why their data fit better with their claim than the above alternative interpretation.

Note that the above argument is assuming the fitness effects are due to secondary mutations, thus the authors' discovery of potential direct effects of mutator-generating mutations (*e.g.mutS* in Figure 1—figure supplement 4) is not affected. However, in the current wording the potential direct benefit of these mutations does not seem to be the main message they try to make impact. It is also not clear why the authors claim "…these direct effects…may not directly influence early selection of hypermutation, but may become decisive in its lower cost at later stages."

---

## [Author Response]

*Reviewer #1:*

*This review is from the point of view of a "general audience" member who is informally interested in mutators. This paper showed that when E. coli was subjected to lethal ethanol stress, mutators evolved in multiple lines. Those mutator lines (but not lines with normal mutation rates) could tolerate lethal levels of ethanol. Mutation rates in these mutator lines were then reduced in later stages of evolution.*

*Authors claim that "an organism's mutation rate is often considered to be a slowly-evolving parameter." I am not sure whether this claim is true since Wielgoss et al., 2013 showed that a hyper-mutator evolved in one of the 12 Lenski lines and that it was later replaced by milder mutators. The difference is that in this study, reduction of mutation rate is through compensatory mutations instead of replacement of one type of mutator with another type. Hammerstrom et al. (2015) (not cited) showed that hypermutators can repeatedly evolve in response to tigecycline (via transposon insertion). Thus, repeated evolution of hypermutator (in contrast to the stochastic appearance of mutator in one of Lenski lines) is not new either. However, there are nice experiments in this paper, and the dynamics of mutation rate is also tracked with fine resolution.*

We thank the reviewer for the appreciation of our work and the detailed revision and suggestions that were made to further improve the manuscript.

Wielgoss *et al.*, 2013 indeed reported the occurrence of a hypermutator and a later decrease in mutation rate in one of the Lenski lines. However, in the light of their work, we still believe our statement of mutation rate being a slowly-evolving parameter is true as this increase and subsequent decrease happened over a long period of time ( ± 40K generations or 20 years) and was observed in only one replicate population. In addition, as also pointed out by reviewer 3, much lower selection coefficients are thought to drive selection on genotypes with lower mutation rates and were observed in studies without stress or under mild stress conditions (Wielgoss *et al.*, 2013; Sniegowski, Gerrish and Lenski, 1997; Barrick *et al.*, 2009; Sandberg *et al.*, 2014). Altogether, we believe that it is valid to state that an organism’s mutation rate is often considered to be a slowly evolving parameter.

Thank you for pointing out the Hammerstrom *et al.* study that shows repeated evolution of *A. baumannii* hypermutators in response to tigecycline treatment (Hammerstrom *et al.*, 2015). It demonstrates that our observation of hypermutation in all high ethanol tolerant lines is not limited to near-lethal ethanol stress, but underlies a broader phenomenon where cells exposed to extreme stress (including ethanol or antibiotics) favor higher mutation rates to avoid extinction. As this excellent work substantiates the clinical relevance of our study, we have cited reference 36 and discussed it both in the Introduction and Discussion sections of the revised manuscript.

However, we believe that our results go far beyond the study of Hammerstrom *et al.* We not only describe rapid, repeated evolution of hypermutation in each high ethanol tolerant line, but also the evolution of multiple, consecutive increases and decreases of the mutation rates within the same evolutionary line. To our knowledge, such a highly dynamic change of mutation rate over time has never been observed before. To stress the difference between the repeated evolution of hypermutation in all high ethanol tolerant lines and the recurrent increases and decreases observed within a line, we changed the Abstract to emphasize the major findings of our study.

“[…]In contrast, we demonstrate an unexpected flexibility in cellular mutation rates as a response to changes in selective pressure. We show that hypermutation independently evolves when different *Escherichia coli* cultures adapt to high ethanol stress. Furthermore, hypermutator states are transitory and repeatedly alternate with decreases in mutation rate. Specifically, mutation rates rise when cells experience higher stress and decline again once cells are adapted.[…]”

*Main points:*

*Subsection “Long-term adaption to high ethanol stree in E. coli is contingent upon hypermutation”: "evolutionary dead ends" – how do you know that? Have you introduced mutator mutations into dead-end lines and observed lack of improved ethanol survival? If so, it will be interesting to know why the "dead-end" lines failed to evolve mutator phenotypes.*

We did not introduce a mutator mutation back into these dead-end lines. Therefore, we have removed and rephrased the term “evolutionary dead-ends” to describe these low ethanol tolerant lines in the manuscript.

“[…]Although ethanol tolerance increased in all populations, only eight out of 20 lines developed tolerance to very high (7% or more) ethanol concentrations (Figure 2), while the other 12 lines recurrently died out and only developed tolerance to relatively low ethanol concentrations (6% or lower).[…]”

“[…]If the strain needed more than 14 days to grow, we assumed this population died out.[…]”

“[…]If the strain did not show growth in a 14-days timespan, we assumed that the line died out and we revived the previous stored intermediate point to restart the evolution.[…]”

While we have no direct evidence, introducing a mutator mutation might possibly allow these lines to further adapt to higher stress levels. To have a better understanding of why these lines did not evolve a mutator phenotype, we sequenced 2 of the low tolerant lines (E4 & E17). Both of these lines accumulated a mutation in *fabA*, a gene involved in fatty acid biosynthesis that has already been linked to higher ethanol tolerance (Dombek and Ingram, 1984; Luo, *et al.*, 2009). However, line E4 additionally contained a mutation in *rpoZ* and line E17 accumulated an additional deletion of *infB*. *rpoZ*encodes a subunit of RNA polymerase and plays a role in transcription, while *infB* encodes a protein chain initiation factor and plays a role in translation. It was recently shown that ethanol both affects transcription and translation in the cell (Haft, *et al.*, 2014). While speculative, the effect of these specific mutations might severely impair growth of these low-tolerant lines on higher percentages of ethanol preventing the occurrence of mutator mutations or others and thus further evolutionary improvement. We have included this in the Discussion section:

“[…]Even though we mainly focused on increased mutation rates in this study, the 12 slowly-mutating, low ethanol tolerant lines might be an interesting starting point for further research. The lack of hypermutation in these lines seems to impede further adaptation to high ethanol concentrations. Sequence analysis of two of these lines revealed the presence of mutations in *rpoZ* (subunit of the RNA polymerase) and *infB* (protein chain initiation factor) in lines E4 and E17, respectively. Since ethanol is toxic through its effect on transcription and translation (Haft, *et al.*, 2014), disruption of the transcription or translation machinery due to these mutations might cause increased sensitivity to higher ethanol levels. This would prevent further growth and the possibility of acquiring a mutator allele or any other mutation. Although we have no evidence supporting that the *rpoZ* and *infB* mutations are causal for the decreased mutation rate, these mutations are interesting and might explain the lack in further adaptive improvement in lines E4 and E17 (Figure 3).[…]”

*Quantification method of ethanol tolerance needs to be described. Figure 5: Units? I am in fact not sure about the point of Figure 5 given that high ethanol tolerance can be achieved with low mutation rate at later stages of evolution.*

Ethanol tolerance of a population was measured as the ability to grow in liquid medium in the presence of a certain percentage of ethanol. When the population grew and reached an optical density of 0.2 or more in the presence of a certain percentage of ethanol it was considered tolerant to that percentage. To ensure the proper percentages of ethanol were added to the medium, we used the Alcolyser beer analyzing system (Anton Paar GmbH, Austria). We now clarified the quantification of ethanol tolerance and measurement of ethanol concentration in the Materials and methods section.

“[…]To ensure the proper percentages of ethanol added to the medium we used the Alcolyser beer analyzing system (Anton Paar GmbH, Austria).[…]”

“[…]Ethanol tolerance of a population was measured as the ability to grow in liquid medium in the presence of a certain percentage of ethanol to an optical density of at least 0.2.[…]”

We have also added units to Figure 5.

Data shown in Figure 5 prove that the dynamics in mutation rate underlie the evolution of ethanol tolerance. There is no correlation between the absolute mutation rate in each point and the absolute ethanol tolerance in each point. Instead, this figure shows that increases in mutation rate between two consecutive points are significantly correlated with increases in ethanol tolerance between those two time points, indicating that a mutator mutation (that causes the increase in mutation rate) hitchhikes with a beneficial mutation (that causes increased ethanol tolerance) and that these beneficial mutations have higher probability to occur in a mutator background. Moreover, decreases in mutation rate (negative Δ mutation rate) are correlated with periods of constant ethanol pressure (Δ ethanol tolerance of zero), showing that mutation rate decreases again once a population is able to grow in the presence of a certain ethanol percentage. This correlation information is lacking from Figure 5, showing the relevance of the extra panel.

*I do not find increased death rate associated with mutators particularly interesting, without authors explaining to me the fundamental difference between slower birth, increased death, or a combination of the two. After all, one can view birth and death as "net-growth".*

We agree that the growth of a population depends on both the birth and the death of the cells in the population. Therefore, we added some lines to the Results section to clarify that our observations are linked to increased death rather than to slower birth.

“[…]Surprisingly, all tested intermediate points showed a very fast decrease in viable cell count when entering the stationary phase (Figure 6—figure supplement 1). This decrease in viable cells is explained by a genuine increased death rate in the population since cells are in stationary phase.[…]”

Also, by plating growing populations at different time points, we were able to observe a decrease in viable cell (i.e. negative net-growth) starting from early stationary phase. Because birth of cells is already absent or low in stationary phase, the decrease in viable cells is explained by increased death of the cells. Moreover, we were able to confirm an increased death by life- dead staining (Figure 6; Figure 6—figure supplement 1; Figure 6—figure supplement 2).

*I do not understand the point of Table 1. Why not recalculate estimated number of mutations based on the dynamics of mutation rate and see whether it matches observed number of mutations?*

Table 1 was meant to show the discrepancy between the expected number of mutations based on the endpoint mutation rate and the actual number of fixed mutations effectively observed in the genomic sequence of the endpoints. It is an interesting suggestion to recalculate the number of mutations based on the dynamics in mutation rate. We tried to calculate the area under the curve (AUC) in Figure 5 to generate an estimated number of mutations over a period of time, taking into account the fluctuations in mutation rate at each time point. However, this calculation is oversimplified. The genomic mutation rate, measured by fluctuation assays, is not suitable for the calculation of estimates that will be compared to the true number of fixed mutations. The genomic mutation rate only reflects the number of mutations occurring in the population, but it does not give an estimate of the number of mutations that will eventually fix in the population. Generating a proper approximation of the expected number of mutations would require estimates of the beneficial mutation rate and the fixation probability of a mutation in a given environment. It would be extremely difficult to infer this in our specific setup as we for example, did not use a constant environment but instead continuously increased the percentage of ethanol. Also, calculations of the necessary parameters to generate a valid estimate would vary for each time point, making it extremely complex. Therefore, we decided to remove the column with the estimated number of mutations and replace the table by an additional supplemental figure, showing both the total number of variants in the population and the number of fixed mutations (Figure 2—figure supplement 2). This additional figure nicely shows that the mutational profile of mutator populations is complex with several low frequency mutations that belong to different subpopulations, possibly having higher or lower mutation rates than the average population mutation rate.

*It will be interesting to know which mutations directly promote ethanol tolerance (if you already have the data).*

The identification of mutations that directly promote ethanol tolerance is extremely challenging in our dataset, given the likely high number of passenger mutations among few driver mutations. However, we are currently developing a method to analyze such highly complex mutational datasets, often resulting from hypermutator evolution. By looking at parallelism between evolved lines we were already able to identify the fatty-acid biosynthesis pathway, encoded by the fab genes, as a main target in the initial adaptation to high ethanol stress. The identification of this pathway serves as a true positive for our analysis since changing the fatty acid composition of the membrane was previously reported as a prime mechanism to promote ethanol tolerance (Dombek and Ingram, 1984; Luo, *et al.*, 2009).

*Reviewer #2:*

*In their manuscript entitled "Adaptive tuning of mutation rate allows fast response to lethal stress in E. coli", Toon Swings and collaborators study the evolution of mutation rate under adaptation to ethanol. They show that increased mutation rate is rapidly selected for in the early stages of the adaptation, and more surprisingly they uncover a rapid decay of mutation rate that seems to be directly linked to some increased mortality. The data are solid, quantitative and very interesting, especially the connection between mortality and mutation rates. My comments are mostly on some if the interpretations of the results, and on some experimental details*

*In the Introduction, it is not fully clear if Ethanol 5% is a real lethal stress: mutations are required to survive the first growth cycle. Based on the lag time is seems so. If it is that case the paper would benefit from referring to the filed of rescue experiments. These are fashionable experiments that study adaptation in conditions in which it is essential for survival. A paragraph (or even the title) on the link between the experiment and that field would increase the readership of the paper.*

We agree with the reviewer and have included the concept of evolutionary rescue and the link with hypermutation in the Introduction and Discussion sections.

“[…]This is especially obvious in harsh environments, where near-lethal stress requires swift adaptation of at least some individuals to avoid complete extinction of the population (Bell and Gonzalez, 2011). Adaptation sufficiently rapid to save a population from extinction leads to so-called evolutionary rescue. This phenomenon is widely studied in the light of climate change and adaptation of declining populations to new, changing environments (Lindsey, *et al.*, 2013). It occurs when a population under stress lacks sufficient phenotypic plasticity and can only avoid extinction through genetic change (Gonzalez, *et al.*, 2013). Evolutionary rescue depends on different factors such as the population size, genome size, mutation rate, degree of environmental change and history of the stress (Gonzalez, *et al.*, 2013; Gonzalez and Bell, 2012). By increasing the supply of mutations, hypermutation might also be crucial to enable evolutionary rescue for populations under lethal stress.[…]”

“[…]Despite the burden of additional, possibly lethal mutations, the increased mutational supply enables fast adaptation of at least some individuals and rescues the population from extinction (4; 46; 26; 25).[…]”

Evolutionary rescue is indeed an interesting phenomenon that is highly relevant for adaptation to lethal stress. 5% ethanol almost completely inhibits growth of the wild type and drastically reduces the carrying capacity (Figure 1—figure supplement 2). The different lag times observed in both the wild type and the mutator mutants indeed suggest that mutations are necessary to enable growth in the presence of 5% ethanol, which would consequently imply that it is a real lethal stress. However, to certify this claim we would need to sequence the strain after growth on 5% ethanol to see if we can actually identify the adaptive mutations that enabled growth and rescued the population from extinction.

Of note, previous studies pointed out that increasing ethanol tolerance is complex, involves interaction between multiple genes and pathways and requires multiple mutations (Goodarzi, *et al.*, 2010; Nicolaou, *et al.*, 2012; Voordeckers, *et al.*, 2015). The need for multiple mutations to achieve higher ethanol tolerance combined with the severely impaired growth under 5% ethanol drastically limits the possibility to acquire adaptive mutations. Therefore, we believe that increasing the mutation rate is the only possibility to generate enough genetic variation in time to enable adaptation and avoid extinction. Although evolutionary rescue largely depends on the mutation rate (Gonzalez, *et al.*, 2013; Gonzalez and Bell, 2012), the occurrence of hypermutation in relation to evolutionary rescue has not been discussed in detail. Possibly because hypermutation-enabled evolutionary rescue requires two mutational events: the occurrence of the mutator mutation and the occurrence of an adaptive mutation that rescues the population. The occurrence of hypermutation in all our high ethanol tolerant lines and the recurrent increases and decreases in mutation rate that match the changes in ethanol pressure during evolution demonstrate that hypermutation is crucial to allow fast adaptation of at least some individuals and rescue the population from extinction under near-lethal stress conditions.

*The paper is clearly showing a new factor that may explain changes in mutation rate: mortality. However, the whole interpretation of the phenomenon is still linked to the idea of second order selection and load. This is really in line with the first theoretical data on mutators and second order selection that neglected all potential direct effects. More recent work have put more emphasis on potential direct effects and I think the discussion should introduce some of these ideas and eventually challenge them. Some results here suggest that eventually some direct effect of mutators may be at play, not in the early selection of mutators but eventually in their costs.*

*First in Figure 4 even at extremely low frequencies, mutator alleles are selected for. This is quite surprising as at some point mutator subpopulation should be to low to generate any beneficial mutations. Indeed if we assume a 1000 fold effect on transitions, mutator should be outcompeted by wild type at a ratio of 1/10000 or so. So Figure 4 suggest a direct effect. Also mentioned in the text the fact that only point mutations were found in mutS and no inactivation is also an argument for a more complex effect than just increase in mutation rate. (inactivation should be much more likely to occur (though some deletions may not be detected at population level with the pipeline used) and as they have an impressive benefit in competition they should invade immediately…*

*Even in the very nice experiments linking mortality and mutation rate, a 10 fold reduction in mutation rate from IM2 to IM3 lead to a two fold decrease in mortality or less in dead cells. So mutation rate is not the only player. Mutation could have accumulated that lead to the mild differences between IM2 and IM3, But in Figure 7, the author found that a three fold change in mutation rate has this time a drastic effect on survival (more than 10 times)And the growth curves are completely different.*

*It may be extremely difficult to go after mechanistic effects, but discussing that both primary selection and second order selection may be at play will benefit the paper.*

*MMR have been shown to be beneficial because they allow a fast switching of flagella expression though increased recombination, have been found to be toxic when too many mismatches are present in the cell such that they lead to double strand break… and few other recent papers have suggested that evolution of mutation rate may also be linked to direct effect. This does not decrease in any way the value of the experiments done, but is worth mentioning in the discussion.*

We thank the reviewer for this interpretation of the data and for his very interesting suggestion to address the role of possible direct effects of MMR on the selection of hypermutation under high ethanol stress. We have now included an extensive discussion of possible directs effects to the manuscript.

“[…]These data indicate that the advantage of hypermutation under ethanol stress can be attributed mainly to second-order selection, following the beneficial effects of novel mutations relative to possible direct effects of the mutator mutation itself.[…]”

“[…] However, the following elements in our results might also suggest potential direct effects of the mutator mutations. First, mainly point mutations were identified in the mismatch repair genes (Figure 2—figure supplement 3) although inactivation of a gene is more likely to occur, given the high competitive benefit of the Δ*mutS* strain compared to the *mutS*_G100A_ strain in the presence of 5% ethanol (Figure 4—figure supplement 1). This would suggest selection of specific changes in the mechanism of the mismatch repair pathway. Second, both the Δ*mutS* and the *mutS*_G100A_ strains still increase in frequency when competed against the wild type in a ratio of 1:1000 or lower (Figure 4). Given the 10- to 50-fold increased mutation rate, a mutator subpopulation at a ratio of 1:1000 or lower should be too small to have an increased chance of acquiring a beneficial mutation compared to the wild-type subpopulation. These data suggest direct beneficial effects of MMR mutations (Cooper, *et al.*, 2012; Torres-Barceló, *et al.*, 2013; Nowosielska and Marinus, 2007) that combined with second-order selection can explain our observations. In addition, we confirmed that any increase in mutation rate, irrespective of the disrupted cellular system, can confer a selective benefit. Therefore these direct effects, which are usually the result of disruptions in one specific system or even in one gene, may not directly influence early selection of hypermutation, but may be obvious in its cost at later stages. […]”

“[…] Some direct effects of already accumulated mutations might explain the inconsistency between the 2-fold difference in mortality between IM2 and IM3 with a 10-fold difference in mutation rate (Figure 6) and the more than 10-fold difference in mortality between END and END Δ*mutS* with only a 2.5-fold difference in mutation rate (Figure 7). […]”

*Of note, with respect to potential rise of mutators, the experiments with a subset of mutators involved in different mechanisms is clearly a proof that increased mutation rate is selected for. Then the fact that only MMR clones are found and no KO is found suggests that there are eventually selection of more moderate effects or of some specific mechanisms.*

Apart from the possible direct effects as described above that might lead to direct selection of mostly MMR mutators, point mutations in the MMR genes might also be selected because they only moderately increase the mutation rate compared to the mutation rate caused by a full knock- out (Figure 4). This would suggest that moderately increased mutation rates are more beneficial because they have a lower cost at later stages compared to higher mutation rate variants. To elaborate on this point, we added a paragraph on this to our Discussion section:

“[…]Both these observations corroborate the idea that moderate mutators will be more easily selected for, because their benefit is higher than low mutation rate variants and their long-term cost is lower than high mutation rate variants. The identification of mostly point mutations leading to amino acid changes and not to nonsense mutations in the MMR genes during evolution similarly suggests selection for mild increases in the mutation rate (such as shown for the *mutS*_G100A_ mutant). […]”

*Reviewer #3:*

[…]

*We have three main comments:*

*Essential information is missing at several places, which prevents both a basic understanding of the population dynamics during the experiment and a comparison of the present findings with those of previous work on the evolution of mutation rates. First, no information on the effective population size of the evolving populations is given: what culture volumes, which dilutions, what final cell densities were obtained and what was the effective population size (~inoculum size x number of generations per growth cycle)?*

We included all the requested information in the manuscript.

“[…]The culture volume during the evolution experiment was 50 ml and dilution was 100- fold at each transfer[…].”

“[…]The minimal optical density of 0.2 corresponds to a minimal final cell density of 5.4x10^8^ CFU/ml in a volume of 50 ml. For each passage, we consequently transferred 500 µl or at least 2.73x10^8^ CFUs. The average number of generations for each growth cycle, estimated based on the optical density reached upon transfer, is 6.67. This value corresponds to the theoretical value resulting from log_2_(dilution factor (100)) = 6.67. Taking these values together, we can calculate an estimated effective population size of 1.82x10^9^ CFU. […]”

*Second, given the high mortality of cells, how were numbers of generations calculated? Dead cells have also been produced and they should be taken into account for the estimates of the total numbers of generations.*

The number of generations was calculated based on the optical density measured at each intermediate time point. First, we calculated the corresponding number of CFU/ml for the measured optical density using an equation derived from the OD_595_-CFU/ml standard curve. Using this CFU/ml value we calculated the total number of CFUs present in the 50 ml culture volume. Since we diluted 100x at each transfer, we could use the total number of CFUs from the previous time point to calculate the inoculum size. Finally, the increase in number of CFUs was calculated by dividing the final number of CFU by the inoculated number of CFU. The log base 2 of this value gives the number of generations (see subsection “Experimental evolution and Equation 3, Materials and methods).

Although this value is a good approximation of the number of generations as OD measurements also record dead cells, we are aware that this calculated value does not specifically take into account the formation of dead cells. Therefore, we added a paragraph to the manuscript explaining how we estimated the number of generations while mentioning the problem of the unknown death rate in the population.

“[…]The number of viable cells was estimated using optical density (A_595nm_) values. This calculation does not specifically take into account the death rate of the cells. However, since the optical density reflects both living and death cells in the culture, calculating the number of generations using the OD values is more accurate compared to calculations based on CFU counts. Indeed, when using the viable cell count data of IM1, IM2 and IM3 (Figure 5), we found a 1.27-fold ( ± 0.29) underestimation of the number of generations when using the number of viable cells as compared to using OD values. […]”

Figure 6—figure supplement 1 shows that the OD only decreases in late stationary phase, when a large proportion of the population is already dead. Given the fact that we transferred the cultures during the evolution experiment in exponential phase at an OD of at least 0.2, this density likely reflect all cells, both living and death in the culture. Hence, generations calculated from these density values will closely approximate the true number of generations. Nonetheless, to calculate the exact number of generations we would need to measure the death rate of each time point separately and extend the growth model with a death rate component. Given the elaborative nature of such experiments this would be practically unfeasible within a reasonable timeframe.

*Third and perhaps most importantly, it was unclear whether fluctuation tests and genome sequence analyses have been done on individual clones or mixed population samples. As a result, we don't know how to interpret Figure 5 and Table 1: do mutation rates in Figure 5 reflect mutation rates of fixed genotypes or are these a mix of mutator and wild-type mutation rates of a polymorphic population? Similar for the frequency of observed mutations in Table 1: are these based on sequences of random clones picked form these populations or on mixed population DNA? The variation in mutation rate shown in Figure 5 suggests that these reflect population mutation rates, since changes are too small to reflect pure genotype effects if these involve mutator mutants such as mutS (see Figure 4). This distinction is crucial for the possible mechanisms that should be considered for explaining the dynamics in mutation rate: if high mutation-rate genotypes fix in the population, later observed low mutation-rate genotypes must derive from the former genotypes, whereas if they are *not* fixed, later observed low mutation-rate genotypes may derive from ancestral genotypes that were still present in the population when the increase in mutation rate was measured.*

All fluctuation assays were done on mixed population samples. Therefore, one possible way of explaining the rapid decrease in mutation rate is by a shift in the population structure where a subpopulation with wild-type mutation rate that was still present, fixes again in the population. However, sequencing analysis on mixed pools using the low frequency variant caller (CLC Genomics Workbench) showed that all found mismatch repair mutator mutations (shown in Figure 2—figure supplement 2) became 100% fixed in the population and remained fixed even after a decrease in mutation rate was observed. Therefore, low mutation rate genotypes must have emerged from the former high mutation rate genotypes. Consequently, selection on low mutation rate genotypes must drive the rapid decrease in mutation rate observed during periods when ethanol concentrations remained constant. The speed of fixation of a mutant with a lower mutation rate and lower mortality might be enhanced by a lower production of dead cells and consequently faster increase in proportion in the population compared to the population with high mutation rate and high mortality.

In addition, we both sequenced the mixed pool and one selected clone from each pool. Table 1 is now replaced by Figure 2—figure supplement 2 that represents the mutations found in the mixed pool sample. Both the total amount of variants (frequency >10%) in each pool as well as the number of “fixed” (>75% frequency) mutations are shown. We now specifically added mention of population mutation rate in the manuscript to clarify the origin of the mutation rate data. Furthermore, we added a paragraph explaining the relevance for interpretation of the data.

“[…]Consequently, the population mutation rate will reflect the average genomic mutation rate of the entire population, containing different subpopulations that possibly display above- or below-average mutation rates. This may explain the discrepancy between the 20-fold increased endpoint mutation rate of line E1 (Figure 3) and the 10-fold increased clonal mutation rate caused by the *mutS*_G100A_ mutation identified in that same line (Figure 4). Furthermore, these data suggest that mutation rate can vary along with population structure throughout the evolution rather than being a fixed rate after the occurrence and spread of one mutator mutation. […]”

*Our second comment is about the surprising rapid selection of genotypes with lower mutation rates at several times during evolution (Figure 5). As sketched above, to understand whether natural selection has driven these changes (and assuming that they reflect the mutation rate of the population, not of individual clones!), two pieces of information are required. First, the fitness consequences of these changes should be quantified in direct competition experiments between genotypes with high and low mutation rate under the conditions prevalent during evolution. Now only the production of dead cells is compared (Figure 6), but no attempt is made to translate mortality estimates into fitness values, either empirically (by running these competitions) or theoretically (by using growth models to predict these effects). The decline of mutation rate during 150 generations of continues evolution for population E9 (Figure 7), and its correlation with lower production of dead cells, are suggestive, but do not solve the riddle how mortality rates translate into fitness under the selective conditions. Second, fitness estimates of high and low mutation-rate genotypes should then be used to predict the dynamic of decline of the mutation rate, to see if this mechanism may indeed explain the dynamics shown in Figure 5.*

We continuously increased the percentage of ethanol during evolution. Since IM2 and IM3 were able to grow on the same percentage of ethanol (7.5%), it would only make sense to compete these intermediate time points against each other to determine fitness values linked to the mortality. However, the difference in number of generations between IM2 and IM3 exceeds 70 and it is plausible that in this time frame other adaptive mutations, anti-mutator mutations and additional genetic load have accumulated. For that reason, we believe that it is highly unlikely that we can directly link a fitness value from a competition between IM2 and IM3 to the mortality. To address this comment we therefore chose to calculate theoretical fitness consequences, rather than running direct competitions.

Accurate prediction of the effect of mortality on the fitness would necessitate setting up a new growth model including a parameter to account for the effect of the death rate. This would require running new experiments to gather enough data to build and substantiate the new model and possibly even a new collaboration with a more theoretically oriented lab. Therefore, we here used the available data and existing growth models to simulate a simplified competition experiment and calculate fitness value and number of selection rounds needed for a lower mutation rate genotype to fix in a population.

First, we determined the different growth parameters, such as growth rate (GR) and carrying capacity (y0). To this end, we used the time to grow in the presence of 7.5% EtOH and final cell density measurements originating from our evolution experiment. This leads to the following parameters for both IM2 and IM3 (Data tables available upon request).

**IM2****IM3****SGR (h^-1^)**0.09730.392
**Y_0_**0.0050.00363
**Y_M_**0.2920.392

Next, we used these values to run a theoretical competition experiment between IM2 and IM3. Therefore, we used the Gompertz growth model (shown below) to calculate the theoretical number of CFU/ml for both strains after a 24 hour growth cycle (y(24)). Here, we made two assumptions. First, we were not able to derive a lag time (LT) value from the data of our evolution experiment. Therefore, we reasonably assumed equal lag times for IM2 and IM3, which is also suggested by the growth curves made for determination of the death rate (Figure 6—figure supplement 1). Second, we assumed that the individual growth dynamics of each strain equals their respective growth dynamics in direct competition with each other.y(x)=y0+ yM∗e[−e((2.718∗SGRyM)∗(LT−x)+1)]

**IM2****IM3****CFU/ml_(o)_**3.95E+063.95E+06**CFU/ml_(_**_24_**_)_**1.73E+086.23E+08

Next, we used these CFU/ml values to calculate the relative fitness of IM3 (low mutation rate) compared to IM2 (high mutation rate), using the discrete time-recurrence equation that describes the spread of a mutant in a haploid population (Figure 8). For A_start_ and A_end_ we used the proportions of IM2 and IM3 in the population determined by the number of CFU/ml for both strains after 24 hours as calculated in the previous step. This calculation resulted in an average relative fitness benefit of 3.61 for the IM3 strain with a low mutation rate compared to the IM2 strain with a high mutation rate.

Finally, we used this relative fitness to estimate the number of selection rounds necessary for fixation of IM2 in a population of IM3 genotypes. The effective population size in our evolution experiment was in the order of 10^9^ CFUs. If we assume that a mutant occurring in the population is present in a 1 over 10^9^ ratio, then we can calculate the number of selection rounds needed to increase in frequency to 99% of the population, given that this mutation is not lost due to random drift. Therefore, we solved the relative fitness equation for n as shown in Figure 8 using both the relative fitness (3.61) and start (10^-9^) and end (0.99) ratios of the competing strain.

Author response image 1.**DOI:**
http://dx.doi.org/10.7554/eLife.22939.025

Using this formula we calculated n necessary for IM3 to fix in a population of IM2. This resulted in an average of approximately 20 (19.74) selection rounds needed for IM3 to fix in a population with initially only IM2 genotypes. Assuming 6.67 generations per selection round this comes down to an average of approximately 132 generations. Consequently, the theoretical time needed to fix based on the calculated fitness parameters exceeds the actual time of 10 selection rounds or 70 generations observed between intermediate points IM2 and IM3. We added a new supplementary graph to show the comparison between calculated and observed number of selection rounds needed for IM3 to fix (Figure 6—figure supplement 3 and subsection “Cellular mortality is the underlying force driving evolution of mutation rates”). The discrepancy between these two values suggests a possible direct effect of further adaptive mutations or of mutator mutations themselves accumulated during consecutive growth steps between IM2 and IM3, as also suggested by Reviewer #2. These two strains not only differ in their mortality and mutation rate, but also in a further buildup of genetic load and possible mutations that allowed further adaptation to the 7.5% ethanol stress. Therefore, it is difficult to link the theoretical fitness calculations directly to the effect of anti-mutators or mortality alone as other factors are present that might influence these values.

Data shown in Figure 7, Figure 7—figure supplement 1 and Figure 7—figure supplement 2 tackle this problem further. Here, we continuously evolved a hypermutator sample from an intermediate time point to reduce the mutation rate. The reduction in mutation rate was accompanied by a reduction in the generation of dead cells. Through the introduction of the Δ*mutS* allele we now obtained both a low mutation rate variant (END) and a high mutation rate variant (END Δ*mutS*) that only differed in their *mutS* allele (and their mutation rate), but are otherwise isogenic. This allowed to calculate the same parameters as we did for IM2 and IM3, but now the results should only represent the effect of the changes in mutation rate and its consequences on the generation of dead cells. To theoretically simulate our competition experiment we calculated the number of selection rounds needed for the END Δ*mutS* strain to decrease in frequency from 50% to less than 1%. This calculation resulted in an average number of selection rounds of 2.5. Indeed, the direct competition between END and END Δ*mutS* starting from a 1:1 ratio demonstrates that after 2 selection rounds the frequency of the END Δ*mutS* strains was reduced to less than 10% (Figure 7—figure supplement 1). Since both competed strains only differ in their mutation rate and mortality, these data confirm that mutation rate and mortality are crucial factors to explain the fast increase of genotypes with a low mutation rate and mortality when the strain is already adapted. To clarify this part in the manuscript, we included an additional graph indicating that there is no significant difference between a theoretical calculation of selection rounds based on the fitness parameter of the strains and observed number of selection rounds necessary for fixation of the END (low mutation rate) strain in a direct competition experiment (Figure 7—figure supplement 2 and subsection “Cellular mortality is the underlying force driving evolution of mutation rates”).

*Our third comment is on Figure 1. First, why would small inocula lead to much higher yields than large inocula? This is surprising, but remains unexplained. Second, the identity of the growth curves should be better explained. What do the grey lines reflect, individual replicates or mean curves for each mutator strain? The number of lines suggest the former, but the text refers to "the" growth curve for dnaQ, suggesting they reflect mean curves. It may be best to show mean growth curves or fitted growth models per strain, with different colors per strain. How are the different populations (of different size) generated? Do they come from the same 'batch' which are either less or more diluted? Or does the 10^7^ dilution come from a later time point of the smaller dilutions? The physiological state of the cells (eg early exponential phase, later exponential phase) at the moment that they are inoculated is known to affect the subsequently generated growth curves. Or perhaps, can larger population sizes 'soak' the ethanol, and thus effectively decrease the concentration in the environment? (see e.g. doi: 10.1098/rsbl.2012.0569)*

All inocula originated from the same batch: The wild type and tested mutants were grown overnight and subsequently diluted to initiate the growth starting from different inoculum sizes. Therefore, we can exclude possible effects of the physiological state of the cell on the growth dynamics. However, we have two hypotheses on the difference in yield between high and low initial inocula.

First, the time axis is different for the two inoculum sizes. We monitored the growth curves starting from a large initial inoculum size for 24 hours compared to 120 hours when starting with a small initial inoculum size. The question here is whether the initial growth on 5% ethanol requires adaptive mutations. We hypothesized that the effect of adaptive mutations would be mitigated by a large initial population size, which is suggested by the results of the experiment and which means that the majority of the population is able to grow without adaptive mutations. However, we did not sequence any strains after the first growth cycle on 5% ethanol to confirm whether adaptive mutations are necessary or not. In addition, we expect more generations in case of the small initial inoculum size (log_2_(dilution factor:100000)=16.61 generations) compared to the case of a large initial inoculum size (log_2_(100)=6.67 generations). Therefore, a mutant occurring in case of a small initial inoculum size will have more time to manifest, possibly leading to the observed higher yield.

Second, growth of a population in the presence of high stress might be facilitated by soaking the ethanol as suggested by the reviewer. Soaking has been described for *P. aeruginosa* that produce a bacteriocin to eliminate possible competitors. The producer cells often have receptors to translocate and neutralize their own bacteriocin, thereby reducing the overall effect of the toxin (Inglis, *et al.*, 2012). Soaking of the ethanol might also happen in larger populations thereby reducing the effective concentration. Ethanol soaking and degradation can happen through mutations in the alcohol dehydrogenase gene *adhE* and has been linked to higher ethanol tolerance before (Goodarzi, *et al.*, 2010).

We changed the text to address the surprising observation of higher yields of small inocula compared to lower yields of large inocula.

“[…] Surprisingly, large initial populations lead to a lower yield compared to small initial population sizes. The growth from the small inoculum is likely driven by adaptive mutations, while the effect of a beneficial mutation might be mitigated when starting with a large inoculum. Moreover, we expect that a mutant occurring in case of a small initial inoculum size will have more time to manifest (log_2_(*dilution factor* ∶ 100 000) = ± 16.61 generations), compared to the mutant occurring in case of a large initial population size (log_2_(100) = ± 6.67 generations), possibly leading to the observed higher yield[…].”

We changed the caption of the figure to clarify the identity of the gray lines in the revised manuscript. The gray lines represent individual replicates of the mutator strains.

“[…]The blue line and shading represents the sigmoidal fit of the wild-type growth curves (n=3, fit using Gompertz equation with 95% c.i. (shading), see Equation 1 in methods section), while the grey lines represent growth curve of separate replicates for each mutator mutant. […]”

The chosen visualization was inspired by a recent paper (Peters, *et al.*, 2016). Showing all the replicates highlights the similarity or diversity in great detail between the growth curves for large and small inocula, respectively. We tried visualizing the average growth curves as shown in Figure 9, but these graphs did not entirely capture the dispersive nature of the growth curves, even though it is still clear that *mutS*, *mutL*, *uvrD*, *mutH*, *mutT* and *mutM* mutants grow much faster than the wild type when starting from a small initial inoculum size.

Author response image 2.**DOI:**
http://dx.doi.org/10.7554/eLife.22939.026

[Editors' note: further revisions were requested prior to acceptance, as described below.]

*Reviewers #4 and 5:*

*This is a study with complicated design and large amount of work. The authors have addressed (or at least tried) most concerns from the last round of revisions. However, as new reviewers, we have some additional points that need the authors' explanations. The paper could be accepted given satisfying responses.*

*1) "A higher mutation rate is linked to a higher mortality, probably due to the extended buildup of genetic load and increased probability of acquiring a lethal mutation". There could be a more straight-forward explanation: Krasovec et al. (2014 Nature Communications) showed that mutation rate from fluctuation assay inversely correlates with population density, due to cell-cell interactions. For the same experimental evolution line at different intermediate time points, higher mortality could reduce population density-given the same inoculum size for fluctuation assay, which in turn increases mutation rate. This alternative interpretation needs to be addressed in the current experimental system.*

This is an interesting suggestion to explain the observed differences in the mutation rate. However, we believe that it is not entirely applicable to our results. We have convincing data showing that mutator mutations occurred prior to changes in mortality and caused the increase in mutation rate, not the way around (Figure 4). We were not only able to identify these mutations (Figure 2—figure supplement 3) but we also show that an increased mutation rate, caused by these mutations, is necessary to enable adaptation under near-lethal ethanol stress. This was demonstrated both with a constructed set of mutation rate variants (Figure 1) as well as in an experimental evolution experiment (Figure 2 and Figure 3). Therefore, we believe that the increases in mutation rate were caused by genuine mutational changes in DNA repair genes and not as a consequence of an increased death rate and subsequent reduction in population density.

Moreover, we carefully measured the population density while performing the fluctuation assays. This was done to avoid possible population density effects on the mutation rate, which were elegantly demonstrated by the Krašovec *et al.* study. We clarified this in the Materials and methods section.

“[…]To ensure correct comparison of the mutation rates we verified that the population densities at the time of plating on rifampicin did not differ significantly. If this was not the case, the test was repeated. The statistical difference between the population densities was measured using a one-way ANOVA with post-hoc Tukey correction. We found no significant difference for any intermediate point compared to the average density and to the density of the other time points. We therefore avoid possible population density effects (Krašovec, *et al.*, 2014, Nat. Commun.). […]”

The final mean densities for each time point of E5 (Figure 5) are now given in panel A of Figure 10. Calculation of the Pearson’s correlation coefficient between the absolute population density and the corresponding mutation rate for each time point (panel B of Figure 10) demonstrated that the population density did not significantly affect the mutation rate in our study. This is also shown by the dashed line representing the linear fitting through the data points which has a slope not significantly different from zero (P = 0.2318). Next, in case the population density would have influenced the mutation rate in our study, an increase in population density should be correlated with a decrease in mutation rate. However, the Pearson’s correlation coefficient between the relative difference in population density between two consecutive time points and the corresponding difference in mutation rate between those two points (panel C of Figure 10) was not significant (P = 0.511). In conclusion, by carefully performing the fluctuation tests when population densities of the strains were not significantly different, we clearly exclude possible effects of cell density on the mutation rate

Author response image 3.**DOI:**
http://dx.doi.org/10.7554/eLife.22939.027

*2) While the authors interpret the data as evidence to support the idea that cells evolve hypermutation to avoid extinction under near-lethal stress, we should also consider a more straight-forward alternative.*

*In the population, mutants with various mutation rates are generated continuously. When the environment turns stressful, the distribution of fitness effects of mutations accordingly becomes wider and less deleterious-mutation-biased due to the decreased population-average fitness (e.g. Hietpas et al., 2013). As a result, the mutation load of hypermutators relative to WT becomes smaller and hypermutators are more likely to survive or even prosper (if hitchhiked with beneficial mutations). The argument can be reversed when organisms become more adapted in the new environment to explain the decline of hypermutators. In this way, the dynamic pattern of mutation rates can be understood as a passive balance between influx (by random mutations) and outflux (due to accumulated genetic load) of mutators without evoking any active response. It would help if the authors can clarify if and why their data fit better with their claim than the above alternative interpretation.*

We thank the reviewers for pointing out this alternative interpretation of the data. However, we believe that this interpretation at least partially overlaps with our current interpretation of the data.

Our results corroborate the idea that only cells with increased mutation rate can generate enough genetic diversity in a short time period to avoid extinction and enable adaptation under near-lethal ethanol stress conditions (potentially aided by a wider and less deleterious- mutation-biased distribution of fitness effects). However, we do not claim that the occurrence of hypermutation itself is an active process. Therefore, our interpretation is largely in line with the interpretation proposed by the reviewers. Indeed, mutants with variations in mutation rate are continuously generated in a population. Moreover, in changing environments the occurrence of hypermutable variants might be facilitated as more mutations are beneficial due to the decreased average population fitness (Hietpas, *et al.,* 2013, Evolution). As suggested by the reviewers, this could mean that, under such near-lethal stress conditions, the genetic load of hypermutable variants is less deleterious-mutation biased, thereby possibly facilitating the emergence of mutators. Here, we moreover believe that these mutators are crucial to enable adaptation and avoid extinction. Since we observed hypermutation in all evolved high ethanol- tolerant lines, but not in the low ethanol-tolerant lines. We have strong data supporting the idea that only cells with a higher mutation rate can rapidly acquire adaptive mutations to avoid extinction. Additionally, our results also confirm that the death rate and mutation rate of a population are linked. These data suggest that higher mortality becomes the cost of hypermutation when a population is already adapted (potentially resulting from a distribution of fitness effects that became again more narrow and deleterious-mutation-biased). Consequently, mutants with a lower mutation rate and lower mortality will have a higher benefit and take over the population, thereby lowering the overall mutation rate. In conclusion, we state that our interpretation largely overlaps with the one proposed by the reviewers. To clarify this in the manuscript we adjusted the text accordingly.

“[…] In conclusion, even though mutator mutants occur spontaneously in the population, these data suggest that hypermutation is a prerequisite, as a driving force, to adapt to high ethanol levels in such a way that only lines with a higher mutation rate than the wild-type mutation rate are able to evolve high ethanol tolerance (Figure 3). […]”

“[…] The rise of hypermutation during adaptation to near-lethal ethanol stress is linked to the idea of second-order selection as suggested by the growth rate and lag time measured for a collection of mutator mutants under 5% ethanol stress (Figure 1; Figure 1—figure supplement 3) and might be facilitated by a wider and less deleterious-mutation-biased distribution of fitness effects in stressful environments (Hietpas, *et al.*, 2013 Evolution). […]”

*Note that the above argument is assuming the fitness effects are due to secondary mutations, thus the authors' discovery of potential direct effects of mutator-generating mutations (e.g. mutS in Figure 1—figure supplement 4) is not affected. However, in the current wording the potential direct benefit of these mutations does not seem to be the main message they try to make impact. It is also not clear why the authors claim "…these direct effects…may not directly influence early selection of hypermutation, but may become decisive in its lower cost at later stages."*

We acknowledge that the possible direct effects of mutator mutations might play a role, even though our data (Figure 1) strongly support the idea that mutators increase in frequency by second-order selection on linked, beneficial mutations. The direct effects might influence which specific mutator mutation eventually spread, but, for the initial adaptation, the effect of hypermutation (with its indirect effects of linked beneficial mutations) clearly is much more important than the (different) small direct effects of specific mutator mutations. To clarify this part, we changed the Discussion section of the manuscript accordingly:

“[…]Therefore, these direct effects, which are usually the result of disruptions of one specific system or even of one specific gene, may influence which specific mutator mutations eventually spread, but will only have a limited effect on the initial selection of hypermutation compared to the direct effect of linked beneficial mutations. However, at later stages these direct effects possibly affect the fate of hypermutators by lowering the cost of the extended buildup of genetic load. […]”